# TimeDART: A Diffusion Autoregressive Transformer for Self-Supervised Time Series Representation

**Daoyu Wang**[1]  **Mingyue Cheng**[* 1]  **Zhiding Liu**[1]  **Qi Liu**[1]

## Abstract

Self-supervised learning has garnered increasing attention in time series analysis for benefiting various downstream tasks and reducing reliance on labeled data. Despite its effectiveness, existing methods often struggle to comprehensively capture both long-term dynamic evolution and subtle local patterns in a unified manner. In this work, we propose **TimeDART**, a novel self-supervised time series pre-training framework that unifies two powerful generative paradigms to learn more transferable representations. Specifically, we first employ a causal Transformer encoder, accompanied by a patch-based embedding strategy, to model the evolving trends from left to right. Building on this global modeling, we further introduce a denoising diffusion process to capture fine-grained local patterns through forward diffusion and reverse denoising. Finally, we optimize the model in an autoregressive manner. As a result, TimeDART effectively accounts for both global and local sequence features in a coherent way. We conduct extensive experiments on public datasets for time series forecasting and classification. The experimental results demonstrate that TimeDART consistently outperforms previous compared methods, validating the effectiveness of our approach. Our code is available at https://github.com/Melmaphother/TimeDART.

## 1. Introduction

The analysis of time-series data has become increasingly critical across diverse application domains, including healthcare (Cheng et al., 2024), finance (Black & Scholes, 1973) and energy management (Zhou et al., 2024). These scenar-

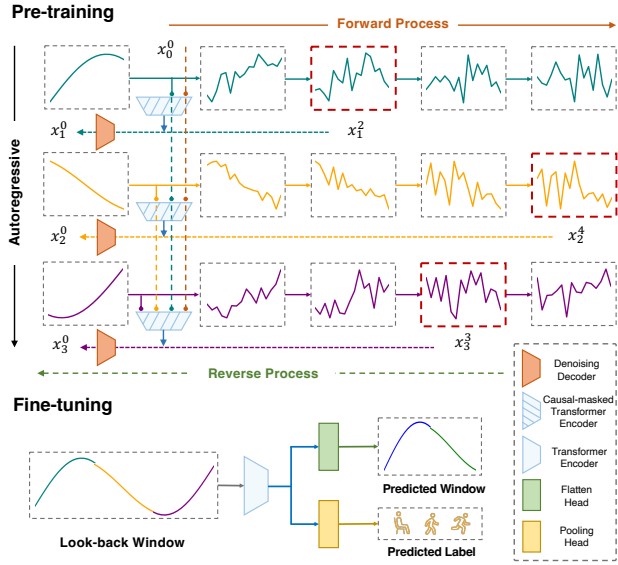

Figure 1: Two-stage training process. Pre-training: TimeDART teaches an encoder to learn temporal representations through a noising process and autoregressive denoising. Fine-tuning: The pre-trained encoder is adapted with task-specific heads for downstream applications.

ios often involve massive amounts of unlabeled real-world data. To analyze and leverage such data, self-supervised learning has emerged as a widely adopted approach (Zhang et al., 2024b). This approach extracts valuable knowledge from abundant unlabeled data, which can be subsequently transferred to boost downstream task performance (Park et al., 2024; Dong et al., 2024b).

Reviewing previous studies (Zhang et al., 2024b), existing methods for time series self-supervised learning primarily fall into three categories. The first category is masked autoencoders, which focus on reconstructing masked or corrupted parts of the input and excel at learning the underlying patterns of the time series (He et al., 2022; Zerveas et al., 2021). A common challenge with these models is the inconsistency between pre-training and fine-tuning due to newly injected masked embeddings (Cheng et al., 2023b). Contrastive-based methods, as the second category, specialize in sequence-level modeling and distinguish between similar and dissimilar time-series segments (Eldele et al.,

[1]State Key Laboratory of Cognitive Intelligence, University of Science and Technology of China, Hefei, China. Correspondence to: Mingyue Cheng <mycheng@ustc.edu.cn>.

*Proceedings of the 42nd International Conference on Machine Learning*, Vancouver, Canada. PMLR 267, 2025. Copyright 2025 by the author(s).

2021; Tonekaboni et al., 2021). Nonetheless, their emphasis on sequence-level modeling may limit their capacity to capture fine-grained temporal variations (Dong et al., 2024a). The third is autoregressive methods, which are well-suited for modeling left-to-right sequential relationships and have promising scale-up potential (Liu et al., 2024c). However, these methods tend to overfit noise and anomalies in time series data (Lim & Zohren, 2021). Moreover, Using the common squared error loss in autoregressive time series modeling tends to make the current value's prediction collapse into a Gaussian distribution centered on past values with a fixed variance (Li et al., 2024b), which deviates significantly from real-world scenarios.

We argue that an ideal self-supervised approach should address the above challenges, that is, how to simultaneously capture long-term dynamic evolution and subtle local patterns in a unified manner. Although the autoregressive self-supervised learning shows significant potential due to its natural alignment with the left-to-right temporal dynamics, a key challenge lies in its limited capacity for modeling fine-grained patterns and its tendency to collapse into a Gaussian distribution. We attempt to address this by incorporating a diffusion denoising process within the autoregressive optimization framework. This design teaches the model to learn useful representations from the explicit added noise during autoregressive pre-training, and rationalizing the squared error of this regression problem using the diffusion loss.

Building upon the analysis above, we propose TimeDART, a novel self-supervised time series representation framework that unifies two powerful generative paradigms, as illustrated in Figure 1. Specifically, we initialize a vanilla Transformer encoder and partition the input time-series data into non-overlapping patches. To prevent information leakage during autoregressive optimization, we apply a causal mask within the encoder. We independently add noise to each patch and use a cross-attention-based denoising decoder, which leverages preceding sequence information to guide the denoising process. TimeDART achieves superior performance across nine publicly available datasets on downstream forecasting and classification tasks.

- We propose TimeDART, a novel self-supervised time series representation framework that unifies autoregressive modeling and denoising diffusion processes to learn more transferable representations.

- We introduce a causal Transformer encoder, coupled with carefully noise addition and denoising, empowering TimeDART to effectively model long-term dynamic evolution and subtle local patterns.

- TimeDART consistently outperforms state-of-the-art baselines across nine public datasets, demonstrating strong adaptability across diverse downstream tasks.

## 2. Related Work

### 2.1. Self-supervised Learning in Time Series

Self-supervised learning has emerged a powerful pre-training approach in fields like natural language processing and computer vision (Brown et al., 2020; He et al., 2022). Unlike supervised learning, which relies on labeled data, self-supervised methods generate supervision from the data's structure. Applying this to time-series data presents unique challenges due to its sequential and temporal characteristics. Current approaches (Zhang et al., 2024b; Cheng et al., 2025a) can be categorized into three main paradigms: masked reconstruction, contrastive discrimination and autoregressive prediction.

**Masked Reconstruction** As a fundamental approach in self-supervised representation learning, mask reconstruction typically involve reconstructing masked or corrupted inputs, encouraging the model to learn meaningful representations. Masked time series modeling, introduced by TST (Zerveas et al., 2021), predicts missing time points from available data. Methods like STEP (Shao et al., 2022), PatchTST (Nie et al., 2023), and CrossTimeNet (Cheng et al., 2025b) extend this approach by operating on sub-series, reducing computational costs while capturing local information. More recent work, such as TimeMAE (Cheng et al., 2023b), introduces decoupled masked autoencoders to address inconsistencies between the pre-training and fine-tuning phases. Additionally, SimMTM (Dong et al., 2024b) improves masked time-series modeling by recovering missing time points through weighted aggregation of neighbors, leveraging the manifold structure of the data.

**Contrastive Discrimination** This approach aims to distinguish between positive and negative instance pairs by pulling similar instances closer and pushing dissimilar ones apart. For example, TNC (Eldele et al., 2021) uses the local smoothness of time series signals to define positive neighborhoods, while TS2Vec (Yue et al., 2022) introduces a hierarchical framework that operates at both the instance and patch levels. Similarly, LaST (Wang et al., 2022) aims to separate seasonal and trend components in time series data within the latent space. CoST (Woo et al., 2022) combines both time and frequency domain information to capture seasonal and trend representations, improving the discriminative power of the learned features.

**Autoregressive Prediction** In time series, autoregressive prediction originated from ARIMA (Box et al., 2015), which combines autoregressive (AR) and moving average (MA) models with differencing. Later, with the rise of RNNs, THOC (Shen et al., 2020) introduced a self-supervised pretext task for multi-resolution single-step forecasting called Temporal Self-Supervision. Recently, inspired by

pre-trained large language models (Brown et al., 2020), the potential of autoregressive models in time series has gained attention, leveraging their generalization ability and task versatility (Liu et al., 2024c). These language models have even been adapted as autoregressive predictors (Liu et al., 2024b) to achieve arbitrary input-output mappings. Nevertheless, the full potential of autoregressive prediction in time series representation learning and various downstream tasks remains to be discovered. To address this gap, TimeDART uses autoregressive optimization during pre-training, improving time series representations and enabling strong performance across downstream tasks.

### 2.2. Diffusion Models in Time Series

In recent years, Denoising Diffusion Probabilistic Models (Ho et al., 2020; Li et al., 2024b) have become powerful tools for time series modeling (Lin et al., 2024; Meijer & Chen, 2024) due to their unique advantages in fine-grained temporal modeling. Early models, such as CSDI (Tashiro et al., 2021), avoid autoregressive inference while incorporating additional input masking. TimeGrad (Rasul et al., 2021) introduced autoregressive denoising with Langevin sampling, enhancing multivariate prediction. More recently, conditional diffusion models like TimeDiff (Shen & Kwok, 2023) have improved time series prediction by leveraging external information to guide the denoising process. However, these methods primarily focus on probabilistic time series prediction tasks, aiming to generate high-quality time series data, but their direct application to downstream tasks like time series forecasting often yields suboptimal results (Zhang et al., 2024a). TimeDART uniquely integrates the denoising diffusion process within a self-supervised framework, enabling inner-patch modeling that preserves the subtle local patterns time series data.

## 3. Methodology

To capture both long-term dynamics evolution and subtle local patterns, TimeDART uses causal masking for autoregressive optimization and a diffusion denoising process to preserve continuous representations. The following sections explore the key components of our approach.

### 3.1. The Proposed TimeDART

TimeDART involves three modules: normalization and patch embedding, causal Transformer encoder and a patch-level diffusion denoising process.

### 3.1.1. NORMALIZATION AND EMBEDDING

**Instance Normalization**   Before feeding the multivariate time series data $\boldsymbol{x}_{1:L} \in \mathbb{R}^{C \times L}$ into the representation network, we apply instance normalization to each instance

$\boldsymbol{x}_{1:L}^{(i)}$, where the superscript $i$ represents the channel index for zero mean and unit variance. After reconstruction, the original mean and standard deviation are restored to maintain input distribution consistency (Kim et al., 2021).

**Patching Embedding**   We use patches as the basic modeling unit, capturing richer local information for more comprehensive representations (Cheng et al., 2023a). Applying noise and denoising to individual points could lead to excessive sensitivity, whereas patches offer more stable embeddings. To preserve the autoregressive property and prevent information leakage, we set the patch length $P$ equal to the stride $S$, ensuring non-overlapping patches that respect the autoregressive assumption. For simplicity, we assume $L$, the time-series length, is divisible by $P$, resulting in $N = \frac{L}{P}$ patches, which reduces computational complexity and facilitates the processing of longer sequences.

Each patch (referred to as a clean patch) is then passed through a linear embedding layer, transforming it into a high-dimensional representation:

$$\boldsymbol{z}_{1:N} = \text{Embedding}(\boldsymbol{x}_{1:N}), \quad (1)$$

where $\boldsymbol{z}_{1:N}$ represents the patch embeddings, and we omit the channel index $(i)$ for simplicity.

### 3.1.2. CAUSAL TRANSFORMER ENCODER

We initialize a vanilla Transformer encoder (Vaswani et al., 2017) as the representation network. During pre-training, we prepend a learnable start-of-sequence (SOS) embedding to the clean patch representations while exclude the final one. To incorporate positional information, we apply sinusoidal positional encoding after the embedding layer. Following this, we use a causal mask $M$ in the processing layer, limiting the visibility of each patch to itself and the prior patches. Finally, the causal encoder network can be expressed as:

$$\boldsymbol{z}_{1:N}^{\text{in}} = \text{Concat}[SOS, \boldsymbol{z}_{1:N-1}] + \text{PE}_{1:N}, \quad (2)$$
$$f(\boldsymbol{z}_{1:N}^{\text{in}}) = \text{Encoder}(\boldsymbol{z}_{1:N}^{\text{in}}, M), \quad (3)$$

where $f(\cdot)$ processes the input sequence with the causal mask, producing the final contextualized representations.

### 3.1.3. PATCH-LEVEL DIFFUSION AND DENOISING

**Forward Process**   For each patch $x_j \in \boldsymbol{x}_{1:N}$, the forward process $q(x_j^s|x_j^{s-1}) = \mathcal{N}(x_j^s; \sqrt{\alpha(s)}x_j^{s-1}, (1 - \alpha(s))I)$ gradually adds noise to the patch, where $\alpha(s)$ is the noise scheduler. Let $\gamma(s)$ be the cumulative product of $\alpha$ over time steps, where $\gamma(s) = \prod_{s' \leq s} \alpha(s')$, the forward process can be rewritten given the original clean patch $x_j^0$:

$$q(x_j^s|x_j^0) = \mathcal{N}(x_j^s; \sqrt{\gamma(s)}x_j^0, (1 - \gamma(s))I). \quad (4)$$

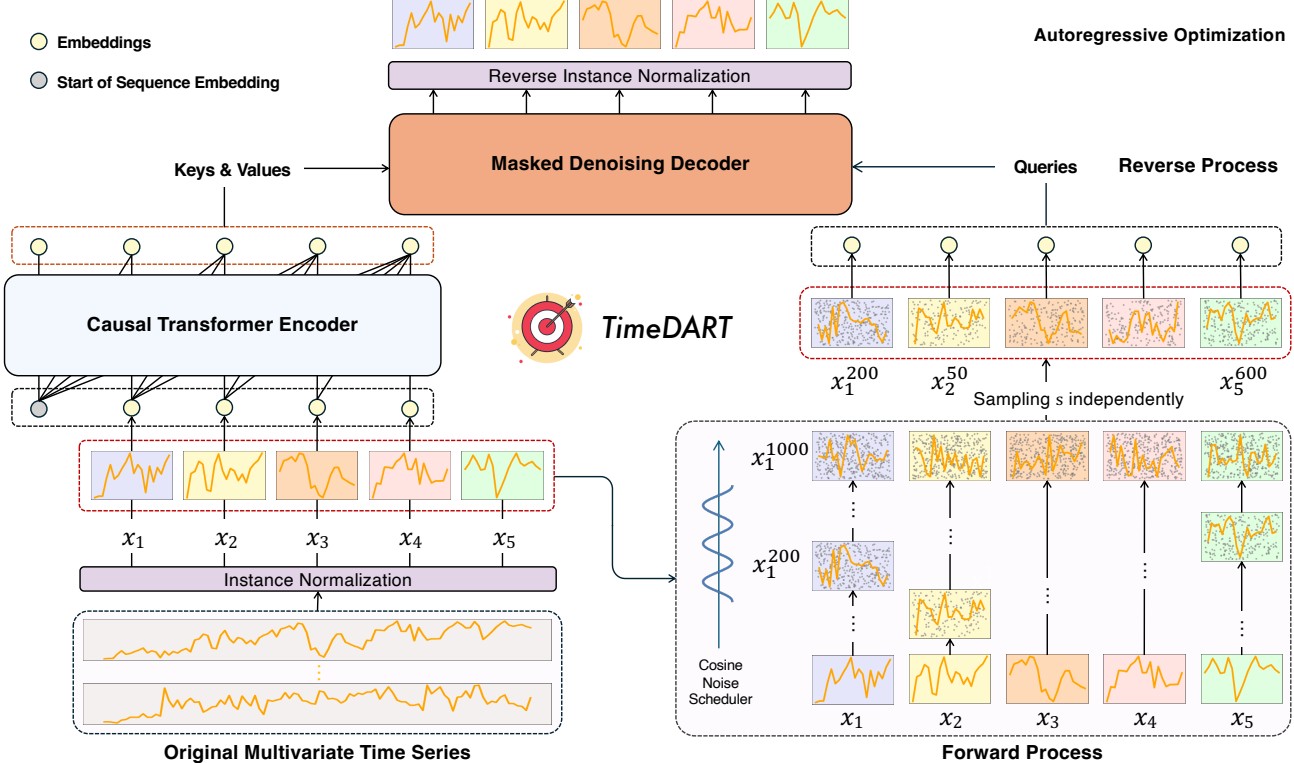

Figure 2: The overall architecture of TimeDART, which employs patch-level modeling, enhances the power of the autoregressive self-supervised method through a diffusion denoising process. It captures both long-term dynamics evolution and subtle local patterns to strengthen consistent and effective representation learning.

As shown in Figure 2, we independently add noise to each patch at time step $s$, enabling the model to learn varying denoising scales across the sequence. This prevents task oversimplification. The noisy patches are represented as:

$$\hat{\boldsymbol{x}}_{1:N} = [x_1^{s_1}, \ldots, x_N^{s_N}]. \tag{5}$$

We use a cosine scheduling approach for the noise scheduler, where $\alpha(s) \propto \cos\left(\frac{s}{T}\pi\right)$, instead of the linear decrease in DDPMs (Ho et al., 2020). This smoother transition emphasizes the early and late stages of diffusion, improving model stability and better capturing the data distribution. Furthermore, both noise-added and clean patches share the same embedding layer and weights, with sinusoidal positional encoding applied:

$$\hat{\boldsymbol{z}}_{1:N}^{\text{in}} = \text{Embedding}(\hat{\boldsymbol{x}}_{1:N}) + \text{PE}_{1:N}, \tag{6}$$

where $\hat{\boldsymbol{z}}_{1:N}^{in}$ denotes embeddings of the noise-added patches.

**Reverse Process**   The reverse process is handled by the denoising decoder which takes the encoder output as keys and values, while the noise-added patch embeddings serve as queries. A self-only mask is applied to the decoder to ensure that the $j$-th input in the noise-added sequence attends only to the $j$-th encoder output. Informed by the causal mask, the

encoder output at position $j$ aggregates information from clean patches at positions 1 to $j-1$, enabling autoregressive optimization. The reverse process is expressed as:

$$z_j^{\text{out}} = g(\hat{z}_j^{\text{in}}, \, f(\boldsymbol{z}_{1:j-1}^{\text{in}})), \quad 1 \leq j \leq N, \tag{7}$$

where $g(\cdot)$ denotes the processing of the encoder output and noise-added patch embeddings by the denoising decoder. Finally, deep representations are mapped back to the original space via linear projection:

$$x_j^{\text{out}} = \text{Projector}(z_j^{\text{out}}). \tag{8}$$

### 3.2. Self-supervised Optimization Objective

Traditional time series autoregressive pre-training objective is typically a simple mean squared error (MSE), aiming to minimize the discrepancy between the regressed and true patches. This can be formalized as follows:

$$\mathcal{L}_{\text{mse}} \propto \sum_{j=1}^{N} ||x_j^0 - \text{Projector}(f(\boldsymbol{z}_{1:j-1}^{in}))||^2. \tag{9}$$

While straightforward, this MSE objective fundamentally assumes that the underlying data distribution is Gaussian.

This assumption is evident when viewing MSE through the lens of maximum likelihood estimation:

$$\frac{1}{2\sigma^2}||x_j^0 - \text{Projector}(f(\boldsymbol{z}_{1:j-1}^{in}))||^2 =$$
$$- \log \mathcal{N}(x_j^0; \text{Projector}(f(\boldsymbol{z}_{1:j-1}^{in})), \sigma^2) + C, \quad (10)$$

where $\sigma$ is a constant and $C$ is a constant determined by $\sigma$. If it is valid, sampling $x_j^0$ could be achieved via $x_j^0 = \text{Projector}(f(\boldsymbol{z}_{1:j-1}^{in})) + \sigma\epsilon$, where $\epsilon$ is standard Gaussian noise. This implies time series observations are either very noisy or follow a unimodal, bell-shaped distribution. Real-world time series, however, often display complex, multimodal behaviors and intricate dependencies that a simple Gaussian assumption poorly captures.

To overcome these limitations, TimeDART replaces MSE with a diffusion loss, allowing the model to express a richer, multimodal belief over time series data. This self-supervised diffusion loss, formally equivalent to the Evidence Lower Bound (Ho et al., 2020; Chen et al., 2024):

$$\mathcal{L}_{\text{diff}} = \sum_{j=1}^{N} \mathbb{E}_{\epsilon, q(x_j^0)} \left[ ||x_j^0 - g(\hat{z}_j^{in}, \, f(\boldsymbol{z}_{1:j-1}^{in}))||^2 \right]. \quad (11)$$

This diffusion loss, embedded in the autoregressive optimization objective, effectively captures both long-term dynamic evolution and subtle local patterns. The core idea is to make the square error a reasonable loss by gradually adding and then removing noise (Su, 2024; Li et al., 2024a). Refer to Appendix C for the detailed derivation.

### 3.3. Downstream Transfering

After pre-training, the denoising decoder is discarded, while the embedding layer and encoder are transferred. The encoder is then adapted with task-specific heads for various downstream tasks. In forecasting, fine-tuning is performed on both the look-back and predicted windows, with a flatten head for one-step prediction, optimized using MSE loss. In classification, fine-tuning is performed on the input sequence with corresponding labels, followed by a max-pooling head that projects the latent representations onto the labels, and is optimized using cross-entropy loss.

## 4. Experiments

We demonstrate TimeDART's effectiveness in forecasting and classification across diverse datasets. Ablation studies confirm the power of its autoregressive mechanism and the critical role of diffusion denoising process, further showing its robustness through parameter analysis.

### 4.1. Experimental Setup

**Datasets**  We summarize the publicly available datasets used in our experiments in Table 1, which includes six

Table 1: Dataset Statistics: The top part shows forecasting datasets, bottom part shows classification datasets.

| DATASETS | DOMAIN | VARIABLES | LENGTH |
|---|---|---|---|
| ETTh1/ETTh2 | Energy | 7 | 17420 |
| ETTm1/ETTm2 | Energy | 7 | 69680 |
| Electricity | Energy | 321 | 26304 |
| Traffic | Transportation | 862 | 17544 |
| Weather | Weather | 21 | 52696 |
| Exchange | Finance | 8 | 7588 |
| PEMS03 | Transportation | 358 | 26208 |
| PEMS04 | Transportation | 307 | 16992 |
| PEMS07 | Transportation | 883 | 28224 |
| PEMS08 | Transportation | 170 | 17856 |
| HAR | HAR | 9 | 14717 |
| Epilepsy | HAR | 3 | 413 |
| EEG | EEG | 2 | 15629 |

datasets for forecasting tasks (both ETT and PEMS datasets contain four subsets each) and three datasets for classification tasks. To clearly illustrate the practical scenarios and scale of these datasets, the table provides an overview of each dataset's domain, the number of samples, and the number of variables per sample. The dataset scale enables us to concretely illustrate the model size and computational requirements of the TimeDART architecture. A more comprehensive description can be found in Appendix A.

**Embedding Layer and Backbone**  In TimeDART, a simple linear transformation is used for the embedding layer. Additionally, to ensure the autoregressive mechanism functions correctly, it is crucial that patches remain non-overlapping to prevent information leakage. Therefore, we prioritize using a vanilla non-overlapping patch embedding layer and a Transformer encoder, rather than relying on existing backbones such as PatchTST (Nie et al., 2023) or iTransformer (Liu et al., 2024a). To investigate whether TimeDART maintains consistent performance across different backbones, we conducted additional experiments by replacing the Transformer encoder with a causal TCN under specific configurations, as it also satisfies the aforementioned requirements. Regardless of the backbone, we use a vanilla Transformer decoder as the denoising decoder, with parameters identical to those of the Transformer encoder, except for the number of layers.

**Baselines**  We compare TimeDART against four state-of-the-art self-supervised baseline methods, including a contrastive learning method: CoST (Woo et al., 2022), and three masked modeling methods: SimMTM (Dong et al., 2024b), TimeMAE (Cheng et al., 2023b) and the self-supervised version of PatchTST (Nie et al., 2023). Since we use a vanilla Transformer encoder as the backbone instead of what is proposed in existing supervised methods, we believe it is essential to compare with advanced supervised methods

Table 2: Multivariate time series forecasting results. All results are averaged MSE and Mean Absolute Error (MAE) from 4 different predicted windows of $\{12, 24, 36, 48\}$ for PEMS datasets and $\{96, 192, 336, 720\}$ for others. The best results are in **bold** and the second best are underlined. Full results are detailed in Appendix D.

| | Ours | | | | Self-supervised | | | | | | | | Supervised | | | |
| | TimeDART | | Random Init. | | SimMTM | | PatchTST | | TimeMAE | | CoST | | PatchTST | | DLinear | |
| Methods / Metric | MSE | MAE | MSE | MAE | MSE | MAE | MSE | MAE | MSE | MAE | MSE | MAE | MSE | MAE | MSE | MAE |
|---|---|---|---|---|---|---|---|---|---|---|---|---|---|---|---|---|
| ETTh1 | 0.411 | **0.426** | 0.439 | 0.444 | **0.409** | 0.428 | 0.433 | 0.437 | 0.434 | 0.445 | 0.465 | 0.464 | 0.427 | 0.435 | 0.439 | 0.449 |
| ETTh2 | **0.346** | **0.387** | 0.358 | 0.396 | 0.353 | 0.390 | 0.354 | 0.393 | 0.402 | 0.431 | 0.399 | 0.427 | 0.357 | 0.395 | 0.458 | 0.459 |
| ETTm1 | 0.344 | **0.379** | 0.351 | 0.383 | 0.348 | 0.385 | **0.342** | 0.380 | 0.350 | 0.383 | 0.356 | 0.385 | 0.362 | 0.388 | 0.361 | 0.383 |
| ETTm2 | **0.257** | **0.316** | 0.269 | 0.323 | 0.263 | 0.320 | 0.272 | 0.327 | 0.270 | 0.326 | 0.282 | 0.343 | 0.270 | 0.329 | 0.281 | 0.343 |
| Electricity | 0.163 | **0.254** | 0.177 | 0.277 | **0.162** | 0.256 | 0.163 | 0.255 | 0.196 | 0.309 | 0.215 | 0.295 | 0.167 | 0.260 | 0.168 | 0.265 |
| Traffic | **0.388** | 0.263 | 0.410 | 0.277 | 0.392 | 0.264 | 0.404 | 0.272 | 0.410 | 0.275 | 0.435 | 0.362 | 0.421 | 0.284 | 0.435 | 0.297 |
| Weather | 0.226 | 0.263 | 0.231 | 0.268 | 0.230 | 0.271 | **0.227** | 0.262 | 0.227 | 0.265 | 0.242 | 0.282 | 0.226 | 0.263 | 0.246 | 0.298 |
| Exchange | **0.359** | **0.405** | 0.440 | 0.450 | 0.451 | 0.455 | 0.376 | 0.413 | 0.427 | 0.446 | 0.456 | 0.455 | 0.379 | 0.414 | 0.393 | 0.425 |
| PEMS03 | **0.152** | **0.257** | 0.164 | 0.266 | 0.158 | 0.260 | 0.156 | 0.261 | 0.165 | 0.269 | 0.169 | 0.273 | 0.178 | 0.288 | 0.277 | 0.373 |
| PEMS04 | **0.133** | **0.245** | 0.145 | 0.255 | 0.143 | 0.253 | 0.139 | 0.249 | 0.144 | 0.256 | 0.147 | 0.262 | 0.149 | 0.266 | 0.290 | 0.381 |
| PEMS07 | **0.128** | **0.232** | 0.138 | 0.243 | 0.131 | 0.236 | 0.132 | 0.237 | 0.137 | 0.241 | 0.139 | 0.245 | 0.149 | 0.253 | 0.322 | 0.387 |
| PEMS08 | **0.201** | **0.282** | 0.213 | 0.293 | 0.206 | 0.286 | 0.206 | 0.287 | 0.211 | 0.292 | 0.215 | 0.295 | 0.230 | 0.295 | 0.359 | 0.402 |

Table 3: Multivariate time series results of general pre-training and fine-tuning across different domains. All results are averaged MSE and MAE from 4 different predicted windows of $\{12, 24, 36, 48\}$ for PEMS datasets and $\{96, 192, 336, 720\}$ for others. The best results are in **bold** and the second best are underlined. Full results are detailed in Appendix D.

| Methods / Domain | Metric | TimeDART | | Random Init. | | SimMTM | | PatchTST | | TimeMAE | | CoST | | In-Domain | |
| | | MSE | MAE | MSE | MAE | MSE | MAE | MSE | MAE | MSE | MAE | MSE | MAE | MSE | MAE |
|---|---|---|---|---|---|---|---|---|---|---|---|---|---|---|---|---|
| Energy | ETTm2 | **0.255** | **0.313** | 0.268 | 0.328 | 0.262 | 0.321 | 0.265 | 0.325 | 0.269 | 0.329 | 0.278 | 0.337 | 0.257 | 0.316 |
| Weather | Weather | **0.227** | **0.263** | 0.238 | 0.272 | 0.233 | 0.268 | 0.229 | 0.263 | 0.229 | 0.263 | 0.254 | 0.284 | 0.226 | 0.263 |
| Finance | Exchange | **0.350** | **0.399** | 0.457 | 0.473 | 0.440 | 0.466 | 0.379 | 0.411 | 0.407 | 0.424 | 0.478 | 0.484 | 0.359 | 0.405 |
| Transportation | PEMS04 | **0.127** | **0.236** | 0.141 | 0.251 | 0.133 | 0.244 | 0.133 | 0.243 | 0.138 | 0.248 | 0.142 | 0.252 | 0.133 | 0.245 |

that feature carefully designed architectures. For the forecasting task, we compare with the supervised version of PatchTST (Nie et al., 2023) and the linear-based method DLinear (Zeng et al., 2023). For the classification task, we compare with FormerTime (Cheng et al., 2023a), which employs a hierarchical Transformer network as the backbone. More implementation details can be found in Appendix B.1.

**Fair Experiment** To ensure experimental fairness, we employ a unified embedding layer and Transformer encoder with consistent model-specific parameters. For forecasting experiments, we implement a channel-independent configuration (Nie et al., 2023) to enhance robustness (Han et al., 2024) and facilitate cross-domain analysis. In addition, we conduct forecasting experiments under two settings: in-domain and cross-domain. In the in-domain setup, a look-back window of $L = 336$ and prediction horizons $H \in \{96, 192, 336, 720\}$ are used for all datasets, except PEMS where $L = 96$ and $H \in \{12, 24, 36, 48\}$. For the cross-domain setting, all forecasting datasets are mixed for general pre-training with a consistent look-back window of $L = 336$, while downstream fine-tuning maintains. To highlight benefits of pre-training, we also include baseline *Random Init.*, involving direct fine-tuning without pre-training. Supervised methods retain their original implementations,

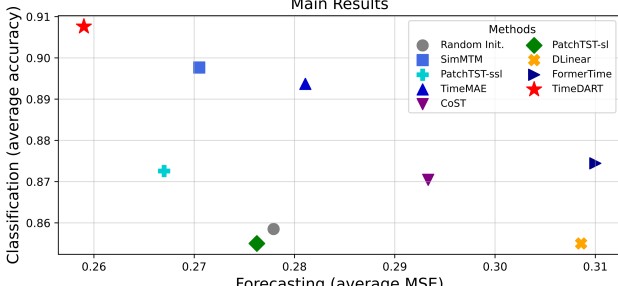

Figure 3: Comparison between TimeDART and baselines for the forecasting task (MSE ↓) across forecasting datasets on the $x$-axis and the classification task (Accuracy ↑) across classification datasets on the $y$-axis.

with only non-core training parameters adjusted for alignment. Further details on experimental fairness are provided in Appendix B.2.

## 4.2. Main Result

As shown in Figure 3, we visually present the performance of TimeDART on two mainstream time series analysis tasks: forecasting (average MSE across forecasting datasets on the $x$-axis) and classification (average accuracy across classification datasets on the $y$-axis). In the majority of experimen-

Table 4: Multivariate time series classification results. Results are are reported as Accuracy (Acc.) and Macro-F1 (F1). The best results are in **bold** and the second best are underlined.

| METHODS | OURS | | | | SELF-SUPERVISED | | | | | | | | SUPERVISED | |
|---|---|---|---|---|---|---|---|---|---|---|---|---|---|---|
| | TIMEDART | | RANDOM INIT. | | SIMMTM | | PATCHTST | | TIMEMAE | | COST | | FORMERTIME | |
| METRIC | ACC. | F1 | ACC. | F1 | ACC. | F1 | ACC. | F1 | ACC. | F1 | ACC. | F1 | ACC. | F1 |
| HAR | **0.9247** | **0.9286** | 0.8738 | 0.8723 | 0.9200 | 0.9220 | 0.8789 | 0.8773 | 0.9204 | 0.9248 | 0.8997 | 0.8927 | 0.8816 | 0.8878 |
| EPILEPSY | **0.9712** | **0.9698** | 0.9265 | 0.9237 | 0.9565 | 0.9543 | 0.9312 | 0.9234 | 0.9459 | 0.9584 | 0.9198 | 0.9156 | 0.9315 | 0.9341 |
| EEG | **0.8269** | 0.5983 | 0.7752 | 0.5138 | 0.8165 | **0.6123** | 0.8076 | 0.5460 | 0.8148 | 0.5787 | 0.7918 | 0.5314 | 0.8102 | 0.5658 |

Table 5: Performance of TCN as backbone. Average MSE and MAE from 4 different predicted windows for forecasting while Accuracy and Macro-F1 for classification task.

| METHOD | TCN | | RANDOM INIT. | | TRANSFORMER | |
|---|---|---|---|---|---|---|
| FORECASTING | MSE | MAE | MSE | MAE | MSE | MAE |
| ETTH2 | 0.349 | 0.396 | 0.357 | 0.403 | **0.346** | **0.387** |
| ETTM2 | 0.263 | 0.323 | 0.269 | 0.326 | **0.257** | **0.316** |
| ELECTRICITY | 0.165 | **0.254** | 0.177 | 0.278 | **0.163** | **0.254** |
| PEMS04 | 0.134 | 0.246 | 0.145 | 0.256 | **0.133** | **0.245** |
| CLASSIFICATION | ACC. | F1 | ACC. | F1 | ACC. | F1 |
| HAR | **0.9252** | **0.9250** | 0.8842 | 0.8901 | 0.9247 | 0.9249 |
| EPILEPSY | **0.9723** | 0.9689 | 0.9525 | 0.9513 | 0.9712 | **0.9698** |

Table 6: The results of ablation study. Average MSE and MAE from 4 different predicted windows for forecasting while Accuracy and Macro-F1 for classification task.

| METHOD | TIMEDART | | w/o AR | | w/o DIFF | | w/o AR-DIFF | |
|---|---|---|---|---|---|---|---|---|
| FORECASTING | MSE | MAE | MSE | MAE | MSE | MAE | MSE | MAE |
| ETTH2 | **0.346** | **0.387** | 0.365 | 0.399 | 0.352 | 0.391 | 0.364 | 0.398 |
| ETTM2 | **0.257** | **0.316** | 0.281 | 0.338 | 0.265 | 0.322 | 0.285 | 0.346 |
| ELECTRICITY | **0.163** | **0.254** | 0.193 | 0.304 | 0.164 | 0.255 | 0.190 | 0.299 |
| PEMS04 | **0.133** | **0.245** | 0.144 | 0.255 | 0.145 | 0.256 | 0.149 | 0.260 |
| CLASSIFICATION | ACC. | F1 | ACC. | F1 | ACC. | F1 | ACC. | F1 |
| HAR | **0.9247** | **0.9286** | 0.8966 | 0.8994 | 0.9002 | 0.9028 | 0.8785 | 0.8756 |
| EPILEPSY | **0.9712** | **0.9698** | 0.9505 | 0.9518 | 0.9598 | 0.9586 | 0.9486 | 0.9472 |

tal settings, TimeDART achieves substantial improvements compared to both self-supervised and supervised methods.

**Forecasting** In the in-domain setting and after downstream fine-tuning for the forecasting task, TimeDART outperforms its competing baselines in most experimental scenarios. As shown in Table 2, TimeDART achieves the best results in 83.3% of the 24 evaluation metrics, with the remaining results being second best. TimeDART further achieves a 6.8% reduction in MSE compared to random initialization and 3% compared to the state-of-the-art benchmarks, despite most methods showing noticeable gains over the random initialized model. Compared to supervised models like PatchTST and DLinear, TimeDART, even with random initialization, consistently performs competitively or better in most cases. Moreover, it significantly surpasses these models after pre-training, highlighting that the use of a vanilla Transformer as the backbone does not limit TimeDART's performance. In fact, this choice underscores the robustness of the proposed pre-training framework. The visualized results are provided in Appendix H to demonstrate the effectiveness of our method.

In the cross-domain setting, after general pre-training, we fine-tune TimeDART on four distinct domains. As shown in Table 3, TimeDART outperforms all competing self-supervised pre-training methods across multiple domains, demonstrating the effectiveness of transferring knowledge from various domains. Moreover, after pre-training, TimeDART significantly outperforms random initialization, highlighting the value of the pre-training process. In most domains, the results after pre-training surpass those

in the in-domain setting, except for the Weather dataset, where performance slightly lags. This may be attributed to the use of the same model-specific parameters across all domains in general pre-training, which could not be finely tuned to the specific requirements of the weather domain. Nevertheless, the results from cross-domain fine-tuning strongly indicate TimeDART's robust capabilities in such scenarios. The full results detailed in Appendix D.

**Classification** The experimental results for the classification task are shown in Table 4. To further explore the effectiveness of TimeDART's representation learning, we also perform supervised fine-tuning on the classification task. The results in Table 4 show that after pre-training, TimeDART outperforms the baselines in terms of accuracy and F1 score in most experimental settings, even surpassing supervised methods specifically designed for classification tasks. Notably, after pre-training, TimeDART improves its average accuracy by approximately 5.7% compared to random initialization. This suggests that through high-level autoregressive optimization and low-level noise-added patch reconstruction, TimeDART captures deep discriminative features of time series data, ensuring its effectiveness in downstream classification tasks.

**TCN as Backbone** To evaluate the effectiveness of TimeDART with different backbones, we replace the Transformer encoder with a causal TCN and remove the causal mask in downstream tasks for consistency. As shown in Table 5, the causal TCN performs slightly worse than the Transformer encoder in forecasting tasks, likely due to the mismatch between the TCN and the Transformer denoising

Table 7: Hyperparameter sensitivity analysis of noise steps and noise schedulers. Average MSE and MAE from 4 different predicted windows for forecasting while Accuracy and Macro-F1 for classification task. See Appendix F.1 for full results.

| ETTm2 | | | | | | PEMS04 | | | | | | Epilepsy | | | | | |
|---|---|---|---|---|---|---|---|---|---|---|---|---|---|---|---|---|---|
| (A) Total Noise Steps | | | (B) Noise Scheduler | | | (A) Total Noise Steps | | | (B) Noise Scheduler | | | (A) Total Noise Steps | | | (B) Noise Scheduler | | |
| Value | MSE | MAE | Type | MSE | MAE | Value | MSE | MAE | Type | MSE | MAE | Value | ACC. | F1 | Type | ACC. | F1 |
| 750 | 0.258 | 0.317 | Cos. | **0.257** | **0.316** | 750 | 0.134 | 0.245 | Cos. | **0.133** | **0.245** | 750 | 0.9195 | 0.9156 | Cos. | **0.9247** | **0.9286** |
| 1000 | **0.257** | 0.316 | Lin. | 0.266 | 0.320 | 1000 | **0.133** | 0.245 | Lin. | 0.143 | 0.254 | 1000 | 0.9247 | **0.9286** | Lin. | 0.8976 | 0.8959 |
| 1250 | **0.257** | **0.315** | Random Init. | 0.269 | 0.323 | 1250 | 0.135 | 0.247 | Random Init. | 0.145 | 0.255 | 1250 | **0.9258** | 0.9279 | Random Init. | 0.9265 | 0.9237 |

Figure 4: Masking strategies: causal mask (left) sees all prior patches, partial causal mask (middle) sees some based on the mask ratio, and self-only mask (right) sees only itself.

Figure 5: Hyperparameter analysis of number layers of denoising decoder and patch length in TimeDART. The triangle symbol represents the best result.

decoder used in pre-training. However, in the classification task, TCN performs competitively, even outperforming the randomly initialized Transformer encoder. This suggests TCN's stronger ability to capture discriminative time-series features. Crucially, TCN still surpasses its random initialization, demonstrating TimeDART's adaptable self-supervised pre-training across diverse backbones.

### 4.3. Ablation Study

We investigate the effectiveness of two key modules: autoregressive mechanism and the denoising diffusion process. Four experimental settings are considered: TimeDART (the original model), $w/o$ $AR$ (autoregressive mechanism removed), $w/o$ $diff$ (denoising diffusion process removed), and $w/o$ $AR$-$diff$ (both modules removed). Specifically, in the autoregressive removal experiment, we eliminate both the causal mask in the Transformer encoder and the mask in the denoising decoder. The results in Table 6 demonstrate that both the autoregressive generation and the denoising diffusion model play crucial roles in the effectiveness of this approach. Notably, removing the autoregressive mechanism results in performance even worse than random initialization, demonstrating its crucial role in capturing long-term dynamic evolution. Similarly, removing the denoising diffusion process also leads to significant performance degradation, highlighting its effective contribution to capture subtle local pattern information.

### 4.4. Hyperparameter Sensitivity

**Forward Process** We first investigate two key parameters through the forward process: the total number of diffusion steps $T \in \{750, 1000, 1250\}$ and the noise scheduler $\alpha(s)$,

comparing cosine and linear schedulers. The number of diffusion steps, with higher $T$ values increasing pre-training difficulty, and the noise scheduler, with the cosine scheduler offering smoother transitions than the linear one, both influence the recovery of clean patches. As shown in Table 7, the total number of noise steps does not significantly impact the difficulty of pre-training. However, the cosine noise scheduler performs substantially better than the linear scheduler while the latter sometimes even yields worse results than random initialization. This highlights the critical importance of the noise scheduler, as insufficiently smooth noise addition can result in significantly poorer outcomes.

**Reverse Process** We then evaluate the impact of two parameters through the reverse process: the layer numbers of the denoising decoder and the patch length. The layer numbers, selected from $[0, 1, 2, 3]$, reflects the relative size of the denoising decoder compared to the fixed 2-layer encoder, with 0 layers serving as the ablation case $w/o$ $diff$. As shown in Figure 5, allocating too many layers to the denoising decoder can result in under-training of the representation network due to an over-concentration of parameters in the denoising component. Additionally, we explore the impact of different mask strategies applied to the denoising decoder, as shown in Figure 4. We find that the self-only mask already contains all the necessary information for denoising, and adding extra information actually hinders the denoising process. These results are detailed in Appendix F.2.

On the other hand, the patch length controls the amount of local segment information that each patch carries. Longer patch length is more effective for datasets like Electricity, which exhibit higher redundancy between consecutive data points, while shorter length is preferred for datasets with less

Table 8: Fine-tuning under few-shot and linear probing settings. Average MSE and MAE from 4 different predicted windows for forecasting while Accuracy and Macro-F1 for classification task. See Appendix G for full results.

| | ETTH2 | | | | | | PEMS04 | | | | | | HAR | | | | |
| | (A) FEW-SHOT | | | (B) LINEAR PROBING | | | (A) FEW-SHOT | | | (B) LINEAR PROBING | | | (A) FEW-SHOT | | | (B) LINEAR PROBING | |
| PORTION | MSE | MAE | METHOD | MSE | MAE | PORTION | MSE | MAE | METHOD | MSE | MAE | PORTION | ACC. | F1 | METHOD | ACC. | F1 |
|---|---|---|---|---|---|---|---|---|---|---|---|---|---|---|---|---|---|
| 5% | 0.365 | 0.392 | TIMEDART | **0.354** | **0.391** | 5% | 0.142 | 0.254 | TIMEDART | **0.145** | **0.258** | 5% | 0.8843 | 0.8901 | TIMEDART | **0.8976** | **0.9005** |
| 10% | 0.354 | 0.391 | SIMMTM | 0.357 | 0.395 | 10% | 0.140 | 0.250 | SIMMTM | 0.148 | 0.260 | 10% | 0.9014 | 0.9053 | SIMMTM | 0.8732 | 0.8756 |
| FULL | **0.346** | **0.387** | TIMEMAE | 0.361 | 0.397 | FULL | **0.133** | **0.245** | TIMEMAE | 0.152 | 0.264 | FULL | **0.9247** | **0.9286** | TIMEMAE | 0.8858 | 0.8862 |
| RAN. FULL | 0.358 | 0.396 | RANDOM INIT. | 0.368 | 0.401 | RAN. FULL | 0.145 | 0.255 | RANDOM INIT. | 0.161 | 0.271 | RAN. FULL | 0.8738 | 0.8723 | RANDOM INIT. | 0.8542 | 0.8578 |

redundancy, as they capture finer temporal dynamics. Figure 5 underscores the importance of adaptive patch length selection based on dataset characteristics.

### 4.5. Analysis Experiment

**Few-shot Evaluation** In real-world applications, labeled time series data often starts with a limited number of observations. This can make it tough to train accurate models. To see how well TimeDART performs in these situations, we conduct few-shot evaluations using the ETTh2, PEMS04, and HAR datasets. Specifically, we train our models using only a small fraction (either 5% or 10%) of the available training data, a method consistent with previous research (Liu et al., 2024d). We then evaluate these models on the complete test set. The results of this evaluation are presented in Table 8 (a).

The results show that while fine-tuning with a small portion of samples yields weaker performance than fine-tuning with the full dataset on both forecasting and classification tasks, it still outperforms full-data supervised fine-tuning under random initialization in the vast majority of cases. This corroborates the strong capabilities of the pre-trained representation network. The results are detailed in Appendix G.

**Linear Probing** As an important fine-tuning setting, we also experiment with the linear probing, where we fix the pre-trained encoder and only fine-tune the newly added projector at the end of the model. Table 8 (b) illustrates the performance compared to some of the compared baselines.

Our findings indicate that TimeDART's performance in the linear probing setup exceed that of several baseline models, providing additional evidence for the efficacy of our pre-training framework. Concurrently, linear probing often matches or surpasses full-data supervised fine-tuning from random initialization (e.g., MSE 0.354 vs. 0.358), demonstrating the generalizability of TimeDART's learned representations across various downstream tasks. The results are detailed in Appendix G.

**Adapt to Extended-Length Input** The standard forecasting framework can often struggle when faced with extended input lengths, a challenge frequently attribute to the increased noise inherent in longer time series. However, TimeDART's pre-training explicitly incorporates noise, ef-

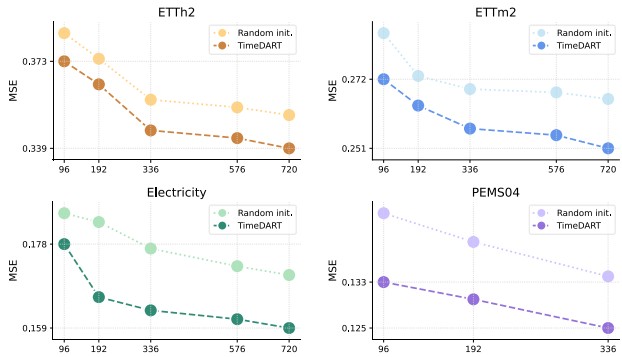

Figure 6: Pre-training and Fine-tuning the model to the inputs with extended length. The MSE averaged from all predicted horizons is reported.

fectively teaching the model to extract valuable representations even from noisy data. This intentional design allows TimeDART to achieve superior forecasting performance with longer look-back windows.

Furthermore, as the look-back window increases, we observe a consistent improvement in forecasting in Figure 6. This demonstrates that TimeDART successfully captures meaningful information from extended input length, and crucially, robustly mitigates the impact of increasing noise that often accompanies longer inputs.

## 5. Conclusion

In this paper, we introduced TimeDART, a novel pre-training framework for self-supervised time series representation learning that unifies two effective generative paradigms. By employing a causal Transformer encoder with a patch-based embedding strategy to model global trends and incorporating a denoising diffusion process to capture fine-grained local patterns, TimeDART effectively addresses both long-term dynamic evolution and subtle local features. Extensive experiments on benchmark datasets for time series forecasting and classification reveal that TimeDART consistently outperforms compared methods, confirming its effectiveness. A series of extra analyses confirmed TimeDART's suitability and robustness in numerous self-supervised settings. With its unique approach, TimeDART has the potential to pave the way for future advancements in self-supervised representation learning for time series.

## Acknowledgments

This research was supported by grants from the National Natural Science Foundation of China (62337001), the grants of Provincial Natural Science Foundation of Anhui Province (No.2408085QF193), the Key Technologies R & D Program of Anhui Province (No. 202423k09020039) and the Fundamental Research Funds for the Central Universities (No. YD2150002501, No. WK2150110032).

## Impact Statement

This paper introduces a novel time series pre-training framework, where we seamlessly unify autoregressive mechanism and the denoising decoder process to learn more transferable representations. Our work offers valuable insights for future research in this domain. Experimental results demonstrate the effectiveness of our method across various domains and its potential real-world applicability. We have ensured that all datasets used in the experiments are publicly available, and we have carefully considered the ethical implications of our work. Therefore, we believe that our research does not present any ethical or moral risks.

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

# A. Dataset Descriptions

To evaluate the effectiveness of our proposed TimeDART, we conducted extensive experiments on 6 time series datasets for forecasting and 3 for classification. These datasets cover a variety of application scenarios, including power systems, transportation networks, and human activity. For detailed descriptions of the datasets and their respective divisions, please refer to Table 9.

Table 9: Full dataset descriptions. *Samples* are organized in (Train/Validation/Test).

| TASKS | DATASETS | LOOK-BACK | PREDICTED | CLASSES | VARIABLES | SAMPLES | DOMAIN | FREQUENCY |
|---|---|---|---|---|---|---|---|---|
| FORECASTING | ETTh1,ETTh2 | 336 | {96,192,336,720} | - | 7 | 8209/2785/2785 | Energy | 1 Hour |
| | ETTm1,ETTm2 | 336 | {96,192,336,720} | - | 7 | 34129/11425/11425 | Energy | 15 Mins |
| | Electricity | 336 | {96,192,336,720} | - | 321 | 17981/2537/5165 | Energy | 1 Hour |
| | Traffic | 336 | {96,192,336,720} | - | 862 | 11849/1661/3413 | Transportation | 1 Hour |
| | Weather | 336 | {96,192,336,720} | - | 21 | 36456/5175/10444 | Weather | 10 Mins |
| | Exchange | 336 | {96,192,336,720} | - | 8 | 4880/665/1422 | Finance | 1 Day |
| | PEMS03 | 96 | {12,24,36,48} | - | 358 | 15724/5241/5243 | Transportation | 5Mins |
| | PEMS04 | 96 | {12,24,36,48} | - | 307 | 10195/3398/3399 | Transportation | 5Mins |
| | PEMS07 | 96 | {12,24,36,48} | - | 883 | 16934/5644/5646 | Transportation | 5Mins |
| | PEMS08 | 96 | {12,24,36,48} | - | 170 | 10713/3571/3572 | Transportation | 5Mins |
| CLASSIFICATION | HAR | 128 | - | 6 | 9 | 8823/2947/2947 | HAR | - |
| | Epilepsy | 206 | - | 4 | 3 | 137/138/138 | HAR | - |
| | EEG | 3000 | - | 8 | 2 | 12787/1421/1421 | EEG | - |

**ETT (4 subsets) (Zhou et al., 2021)**  This dataset comprises time series data of oil temperature and power load collected from electricity transformers spanning July 2016 to July 2018. It is divided into four subsets, each with different recording intervals: ETTh1 and ETTh2 have hourly recordings, while ETTm1 and ETTm2 are recorded every 15 minutes.

**Electricity (UCI, 2015)**  This dataset captures the electricity consumption of 321 clients on an hourly basis from 2012 to 2014, with measurements taken every 15 minutes (in kW). Time stamps follow Portuguese time. Each day includes 96 measurements (24×4), and during time changes in March (where one hour is skipped), the values between 1:00 am and 2:00 am are set to zero. Conversely, in October (with an extra hour), consumption between 1:00 am and 2:00 am represents the aggregated values of two hours.

**Traffic (PeMS, 2001)**  Road occupancy rates, measured hourly, were collected from 862 sensors located along the San Francisco Bay area freeways. The data spans from January 2015 to December 2016.

**Weather (Wetterstation, 2006)**  This dataset contains meteorological time series featuring 21 indicators. The data was collected every 10 minutes in 2020 by the Weather Station at the Max Planck Biogeochemistry Institute.

**Exchange (Guokun Lai, 2017)**  This dataset collects the daily exchange rates of eight countries—Australia, the UK, Canada, Switzerland, China, Japan, New Zealand, and Singapore—from 1990 to 2016.

**PEMS (4 subsets) (PeMS, 2001)**  This dataset includes publicly available traffic network data from California, recorded in 5-minute intervals. We utilize the same four public subsets (PEMS03, PEMS04, PEMS07, PEMS08) as those used in SCINet (Liu et al., 2022).

**HAR (Anguita et al., 2013)**  The Human Activity Recognition (HAR) dataset is a multi-class classification dataset containing wearable sensor data from subjects performing activities such as walking, sitting, standing, and climbing stairs. It provides rich temporal patterns for recognizing human activities.

**Epilepsy (Andrzejak et al., 2001)** The Epileptic Seizure Recognition dataset consists of EEG recordings from 500 subjects, where the brain activity was recorded for each subject for 23.6 seconds. This dataset provides a valuable resource for studying seizure patterns and developing models for reliable seizure detection.

**EEG (Goldberger et al., 2000)** The EEG datasets (as known as Sleep-EDF) contain polysomnographic recordings, including Electroencephalogram (EEG) signals, for multi-class sleep stage classification. They support automatic sleep stage identification and the study of sleep disorders.

## B. Implementation Details

### B.1. Compared Baselines

To more comprehensively evaluate the capabilities of TimeDART, we compare it against the most advanced self-supervised and supervised models. Among the self-supervised methods, we include contrastive learning and mask modeling, which help us validate the strength of TimeDART's pre-training approach. The supervised models are chosen to compare against the advanced backbones proposed in existing supervised methods, ensuring that the use of the vanilla Transformer for autoregressive optimization in TimeDART does not lead to any inherent limitations in backbone selection. This selection of baselines ensures a robust and comprehensive comparison, highlighting the performance of TimeDART in different learning paradigms and validating its efficacy in a wide range of settings.

**SimMTM (Dong et al., 2024b)** SimMTM[1] leverages manifold learning to recover masked time points by aggregating complementary temporal variations from multiple neighbors outside the manifold, improving the masked modeling task for time series pre-training.

**PatchTST (Nie et al., 2023)** PatchTST[2] introduces a channel-independent patching mechanism for Transformer-based time series models, enabling efficient subseries-level embeddings and shared weights across univariate channels, significantly improving long-term forecasting and self-supervised pre-training performance.

**TimeMAE (Cheng et al., 2023b)** TimeMAE[3] introduces a decoupled autoencoder architecture with sub-series partitioning and random masking to enhance contextual representations and encoding efficiency for time series, using bidirectional encoding and masked codeword classification for effective self-supervised learning.

**CoST (Woo et al., 2022)** CoST[4] leverages contrastive learning in both time and frequency domains to disentangle seasonal-trend representations for time series forecasting, enabling robust performance across backbone encoders and achieving state-of-the-art results.

**DLinear (Zeng et al., 2023)** DLinear[5] introduces simple one-layer linear models that bypass the temporal information loss inherent in Transformer-based self-attention, achieving superior performance in long-term time series forecasting across diverse datasets.

**FormerTime (Cheng et al., 2023a)** FormerTime[6] combines hierarchical multi-scale representation learning with a novel Transformer encoder, integrating temporal reduction attention and contextual positional encoding to enhance efficiency and classification performance for multivariate time series.

### B.2. Implementation Details for Fair Experiment

All experiments were implemented using PyTorch (Paszke et al., 2017) and executed on a single NVIDIA RTX 4090 24GB GPU. To ensure the fairness of comparisons, we adopted a unified encoder as the representation network for all

---

[1]https://github.com/thuml/simmtm
[2]https://github.com/yuqinie98/PatchTST
[3]https://github.com/Mingyue-Cheng/TimeMAE
[4]https://github.com/salesforce/CoST
[5]https://github.com/cure-lab/LTSF-Linear
[6]https://github.com/icantnamemyself/FormerTime

self-supervised methods, maintaining consistency in model-specific parameters. For supervised methods, we adhered to their original implementations without altering model-specific parameters, making adjustments only to non-core training parameters for alignment. For forecasting experiments, we applied the channel-independent configuration (Nie et al., 2023), ensuring consistent settings across methods.

**Pre-training**   The representation network consists of 2 layers for most datasets, while for the Traffic dataset, it comprises 3 layers due to its higher complexity. The batch size was set to 16 for all datasets, except for Traffic and PEMS datasets, where it was reduced to 8 due to memory and time limitations. The ADAM optimizer (Kingma & Ba, 2014) was used with initial learning rates selected from $\{10^{-3}, 5 \times 10^{-4}, 10^{-4}\}$, and L2 loss was employed for optimization. The pre-training process spanned 50 epochs. Sequence representation dimensions were chosen from $\{8, 16, 32, 64, 128\}$.

**Downstream Forecasting Task**   For the forecasting task, a consistent look-back window length of $L = 336$ and predicted window lengths $H \in \{96, 192, 336, 720\}$ were applied to all datasets, except for the PEMS datasets, where $L = 96$ and $H \in \{12, 24, 36, 48\}$. Fine-tuning was performed for 10 epochs, with settings largely consistent with pre-training. The channel-independent configuration (Nie et al., 2023) was utilized, ensuring fairness in encoding time series data across all methods. Mean Squared Error (MSE) loss was employed as the loss function for forecasting tasks, ensuring consistency across experiments. To emphasize the contributions of pre-training, we also conducted experiments with *Random Init.*, where the representation network was randomly initialized and directly fine-tuned without pre-training.

**Downstream Classification Task**   For classification tasks, we followed similar settings, employing the same unified encoder and ensuring that all methods used consistent model-specific parameters. Cross-Entropy loss was utilized as the loss function for classification, providing a standard measure for optimizing multi-class predictions. For fair comparison, all experiments adhered to the same training and evaluation protocols, aligning the sequence representation dimensions and training procedures.

## C. Derivation Details

### C.1. Task Definition

**Time Series Self-Supervised Pre-training**   Given a multivariate time series input $\boldsymbol{X} \in \mathbb{R}^{C \times L}$, the goal is to train an encoder to learn feature representations without labels. The encoder learns temporal patterns through self-supervised tasks, producing embeddings $\boldsymbol{Z} \in \mathbb{R}^{L' \times D}$, where $L'$ is the sequence length after encoding, and $D$ is the dimension of the hidden space. These embeddings serve as representations for downstream tasks.

**Time Series Forecasting**   Given a multivariate time series input $\boldsymbol{X} \in \mathbb{R}^{C \times L}$, where $C$ is the number of channels and $L$ is the length of the back window, the goal is to predict future values $\boldsymbol{Y} \in \mathbb{R}^{C \times H}$ over a predicted window $H$. Here, $\boldsymbol{X} = [\boldsymbol{x}_1, \ldots, \boldsymbol{x}_L]$ comprises $L$ input vectors $\boldsymbol{x}_i \in \mathbb{R}^C$, and $\boldsymbol{Y} = [\boldsymbol{y}_{L+1}, \ldots, \boldsymbol{y}_{L+H}]$ represents the predictions. Pre-training is performed on the look-back window, and both windows are used in the forecasting task.

**Time Series Classification**   Given a multivariate time series input $\boldsymbol{X} \in \mathbb{R}^{C \times L}$ and its class label $y$, the goal is to map $\boldsymbol{X}$ to a probability distribution over $y$.

### C.2. Why We Add Noise Independently?

As shown in Figure 2, for $N$ patches in a sequence, we independently add noise to each patch, meaning that each patch undergoes a different noise addition process. This is done to prevent the sequence from retaining invariant statistical properties, such as mean and variance, after the noise addition. We now prove this:

Assume that no patch operation is applied. After the original sequence is input, it passes through an Instance Normalization layer. After normalization, the mean and variance of the original sequence along the seq_len dimension are 0 and 1, respectively. That is:

$$\mathrm{Mean}(x_{1:L}) = 0 \tag{12}$$

$$\mathrm{Var}(x_{1:L}) = 1 \tag{13}$$

The added noise $\epsilon$ follows standard gaussian distribution $\mathcal{N}(0, 1)$. Suppose that we apply the same noise addition step $s_t$ for each point, then the noisy sequence satisfies:

$$\hat{x}_j = \sqrt{\gamma(s_t)}x_j + \sqrt{1 - \gamma(s_t)}\epsilon, \quad \epsilon \sim \mathcal{N}(0, 1) \tag{14}$$

Next, we calculate the mean and variance of the noisy sequence. Since the distribution of the original sequence $x_{1:L}$ is independent of the noise distribution:

$$
\begin{aligned}
\text{Mean}(\hat{x}_{1:L}) &= \sqrt{\gamma(s_t)}\,\text{Mean}(x_{1:L}) + \sqrt{1 - \gamma(s_t)}\,\text{Mean}(\epsilon) \\
&= 0
\end{aligned}
\tag{15}
$$

$$
\begin{aligned}
\text{Var}(\hat{x}_{1:L}) &= (\sqrt{\gamma(s_t)})^2\,\text{Var}(x_{1:L}) + (\sqrt{1 - \gamma(s_t)})^2\,\text{Var}(\epsilon) \\
&= \gamma(s_t) + (1 - \gamma(s_t)) \\
&= 1
\end{aligned}
\tag{16}
$$

Since the patch division is non-overlapping, the process of dividing patches does not affect the overall sequence distribution. Therefore, the above conclusions still apply.

Thus, if the noise addition step is the same for each patch, it will cause the mean and variance of the noisy sequence to remain the same as before, leading to an oversimplification of the pretraining task. Therefore, we adopt an independent noise addition strategy, and the corresponding experimental results can be found in Table 18.

### C.3. Optimization Objective Derivation

The self-supervised optimization objective we employ follows the classical form of diffusion loss, which is designed to maximize the marginal likelihood of the data $p(\boldsymbol{x}_0)$. In this context, we assume that $p$ represents the reverse denoising process, where the model learns to reconstruct the original data $\boldsymbol{x}_0$ from its noisy versions. This denoising process is modeled as a gradual reverse transformation of the corrupted data, recovering the underlying clean distribution. The ideal loss function for this process can be formally expressed as:

$$\mathcal{L}_{\text{ideal}} = \sum_{j=1}^{N} H(p_\theta(x_j^0),\ q(x_j^0)) = \sum_{j=1}^{N} \mathbb{E}_{q(x_j^0)}[-\log p_\theta(x_j^0)] \tag{17}$$

Use the Evidence Lower Bound (ELBO) method and ignore the patch index subscript:

$$L_{\text{ideal}} \leq \mathbb{E}_{q(x^0)}\left(\mathbb{E}_{q(x^{1:T}|x^0)}\left[\log \frac{q(x^{1:T}|x^0)}{p_\theta(x^{0:T})}\right]\right) \tag{18}$$

$$= \mathbb{E}_{q(x^{0:T})}\left[\log \frac{q(x^{1:T}|x^0)}{p_\theta(x^{0:T})}\right] \tag{19}$$

$$:= L_{\text{diff}} \tag{20}$$

Expand and use the Bayes formula:

$$L = \mathbb{E}_{q(x^{0:T})}\left[\log \frac{q(x^{1:T}|x^0)}{p_\theta(x^{0:T})}\right] \tag{21}$$

$$= \mathbb{E}_{q(x^{0:T})}\left[\log \frac{\prod_{s=1}^{T} q(x^s|x^{s-1})}{p(x^T)\prod_{s=1}^{T} p_\theta(x^{s-1}|x^s)}\right] \tag{22}$$

$$= \mathbb{E}_{q(x^{0:T})}\left[\log \frac{q(x^T|x^0)}{p(x^T)} + \sum_{s=2}^{T}\log \frac{q(x^{s-1}|x^s, x^0)}{p_\theta(x^{s-1}|x^s)} - \log p_\theta(x^0|x^1)\right] \tag{23}$$

$$\tag{24}$$

The first term is a constant, the third term is the reconstruction loss, and only the second term is discussed:

$$\mathbb{E}_{q(x^{0:T})} \sum_{s=2}^{T} \log \frac{q(x^{s-1}|x^s, x^0)}{p_\theta(x^{s-1}|x^s)} = \int \mathrm{d}x^{0:T} \, q(x^{0:T}) \cdot \sum_{s=2}^{T} \log \frac{q(x^{s-1}|x^s, x^0)}{p_\theta(x^{s-1}|x^s)} \tag{25}$$

$$= \sum_{s=2}^{T} \mathbb{E}_{q(x^0, x^s)} \left[ D_{KL}(q(x^{s-1}|x^s, x^0) \| p_\theta(x^{s-1}|x^s)) \right] \tag{26}$$

Applying Bayes' theorem again:

$$q(x^{s-1}|x^s, x^0) = \mathcal{N}(x^{s-1}, \tilde{\mu}^s(x^s, x^0), \tilde{\beta}_s I) \tag{27}$$

It can be assumed that the predicted distribution can also be expressed as:

$$p_\theta(x^{s-1}|x^s) = \mathcal{N}(x^{s-1}; \mu_\theta(x^s, s), \sigma_s^2 I) \tag{28}$$

According to the KL divergence of two Gaussian distributions with the same variance, we can get:

$$L_{\mathrm{diff}} = \mathbb{E}_{q(x^0, x^s)} \left[ \frac{1}{2\sigma_s^2} \| \tilde{\mu}_s(x^s, x^0) - \mu_\theta(x^s, s) \| \right] \tag{29}$$

Since $\tilde{\mu}_s$ and $x^0$ are linearly related, we turn to predict the original space, so:

$$L_{\mathrm{diff}} = \mathbb{E}_{\epsilon, q(x^0)} \left[ ||x^0 - x^{out}||^2 \right]. \tag{30}$$

Adding the cumulative sum of subscripts and the notation from the original paper, we get:

$$\mathcal{L}_{\mathrm{diff}} = \sum_{j=1}^{N} \mathbb{E}_{\epsilon, q(x_j^0)} \left[ ||x_j^0 - g(\hat{z}_j^{in}, \, f(z_{1:j-1}^{in}))||^2 \right] \tag{31}$$

## D. Full Result for Forecasting Task

Due to space limitations, the complete tables for the forecasting task are provided in Appendix H. Experiments in the in-domain setting are shown in Tables 10 and 11, while experiments in the cross-domain setting are presented in Table 12. The experiments with the backbone replaced by causal TCN are shown in Table 13.

## E. Ablation Study Full Result

The results in Table 14 underscore the critical roles of both the autoregressive generation and denoising diffusion components in TimeDART. Removing the autoregressive mechanism (*w/o AR*) leads to a significant performance decline, particularly in ETTm2. At the 96-step horizon, MSE increases from 0.165 to 0.184, and at 336 steps, it rises from 0.279 to 0.307. This illustrates the crucial role of the autoregressive mechanism in enhancing the model's forecasting ability, especially across various time horizons. Similarly, eliminating the denoising diffusion module (*w/o Diff*) results in noticeable performance degradation, as observed in ETTh2. At the 96-step horizon, MSE increases from 0.283 to 0.288, and at the 336-step horizon, it rises from 0.365 to 0.372. These findings highlight the essential contribution of the denoising diffusion process to improving the model's learning and overall performance.

When both components are removed (*w/o AR-Diff*), the model's performance deteriorates significantly across all datasets. For instance, in Electricity, at the 336-step horizon, MSE jumps from 0.166 to 0.199, clearly showing the combined importance of both modules for achieving optimal performance.

In summary, both modules are indispensable for TimeDART's success. The autoregressive mechanism is particularly important for long-term predictions, as evidenced in ETTm2, while the denoising diffusion process significantly improves accuracy and learning, especially in datasets like ETTh2.

# F. Hyperparameter Sensitivity

## F.1. In Forward Process

**Noise Steps**    The results in Table 16 suggest that varying the total number of diffusion steps ($T$) has a relatively minor impact on model performance across datasets. Whether $T$ is set to 750, 1000, or 1250, the model's effectiveness remains consistent, with minimal variation in MSE values. This indicates that once a sufficient number of diffusion steps are reached, further increases offer little additional benefit.

**Noise Scheduler**    In contrast, the noise scheduler plays a more critical role in shaping model performance. The cosine scheduler consistently outperforms the linear scheduler, with the gap in performance widening as the prediction horizon increases. For instance, in the ETTh2 dataset, the cosine scheduler shows significantly better results at longer horizons compared to the linear scheduler, highlighting its ability to facilitate smoother noise transitions. These results emphasize the importance of selecting an appropriate noise scheduler, as it greatly influences the model's ability to effectively denoise during pre-training.

**Shared Embedding Layer**    As shown in Figure 2, clean patches and noise-added patches share the embedding layer in the feed-forward encoder and denoising decoder. We believe that sharing the embedding layer weights allows the model to learn the representation of manually added noise, which helps better distinguish the unavoidable noise and anomalies in time series data. In addition, sharing the weights reduces the overall parameter size of the model, speeding up the computation. As shown in Table 18, the experimental results also confirm our hypothesis: sharing weights consistently outperforms using separate embedding layers.

## F.2. In Reverse Process

**Layers of Denoising Decoder**    The results in Table 17 indicate that increasing the number of layers in the denoising decoder does not consistently improve performance. While a single decoder layer generally provides the best balance between model complexity and accuracy, adding more layers tends to offer diminishing returns. In fact, beyond one or two layers, performance gains become negligible, and excessive layers can even hinder the training process by shifting capacity away from the representation network. This suggests that an overly complex decoder may underutilize the model's capacity, leading to suboptimal pre-training outcomes. Overall, the results emphasize the importance of maintaining a balanced architecture, where one decoder layer appears to be sufficient for effective performance across datasets.

**Mask Ratio**    As shown in Figure 4, we conduct experiments on the mask ratio of the denoising decoder. For the denoising decoder, a self-only mask is sufficient to capture all the information under the autoregressive mechanism, as the Transformer encoder output at position $j$ already aggregates information from clean patches at positions 1 to $j - 1$. We investigate whether providing additional information to the denoising decoder benefits pre-training. The results in Table 15 indicate that redundant information is unnecessary for the denoising process, and as the masked portion decreases, model pre-training performance worsens. This suggests that redundant information confuses the model, increasing pre-training difficulty. However, even with the inclusion of a causal mask, the model outperforms random initialization, demonstrating the stability of the autoregressive mechanism.

**Patch Length**    The results in Table 19 demonstrate that patch length significantly affects model performance, with each dataset benefiting from different levels of information density. For instance, datasets like *Electricity*, which exhibit high redundancy between data points, perform best with larger patches (e.g., patch length 8), achieving the lowest average MSE of 0.163 and MAE of 0.254. In contrast, other datasets may require shorter patch lengths to capture more localized patterns. However, using smaller patches increases the computational complexity considerably, making training much more difficult and resource-intensive. Thus, determining the optimal patch length depends not only on the dataset's characteristics but also on the balance between performance and computational feasibility.

# G. Analysis Experiment

**Few-shot Evaluation**    Table 20 demonstrates the full results of our few-shot fine-tuning evaluation. While fine-tuning with a reduced portion (5% or 10%) of samples generally showed weaker performance compared to using the full dataset, TimeDART consistently outperformed full-data supervised fine-tuning under random initialization across all prediction

windows for both forecasting and classification tasks. This comprehensive outperformance further solidifies the strong capabilities of our pre-trained representation network.

**Linear Probing**    Table 21 also illustrates the full performance of TimeDART in the linear probing setup, in comparison to several baseline models. Our findings indicate that TimeDART's linear probing performance consistently exceeded that of the compared baselines across all prediction windows. This provides further evidence for the efficacy and generalizability of our pre-training framework and the learned representations across diverse downstream tasks.

**Memory and Efficiency**    To better illustrate the proposed TimeDART model's computation cost, we've provided quantitative results on the Traffic dataset, using a lookback window of $L = 336$ and a predicted window of $H = 336$, in Table 22. In summary, TimeDART is a medium-sized model compared to our baselines. Thanks to our approach of training only the diffusion model without requiring lengthy sampling, both the pre-training and fine-tuning processes of TimeDART are quite efficient and don't take up too much time, especially when compared to other pre-training methods.

## H. Visualization

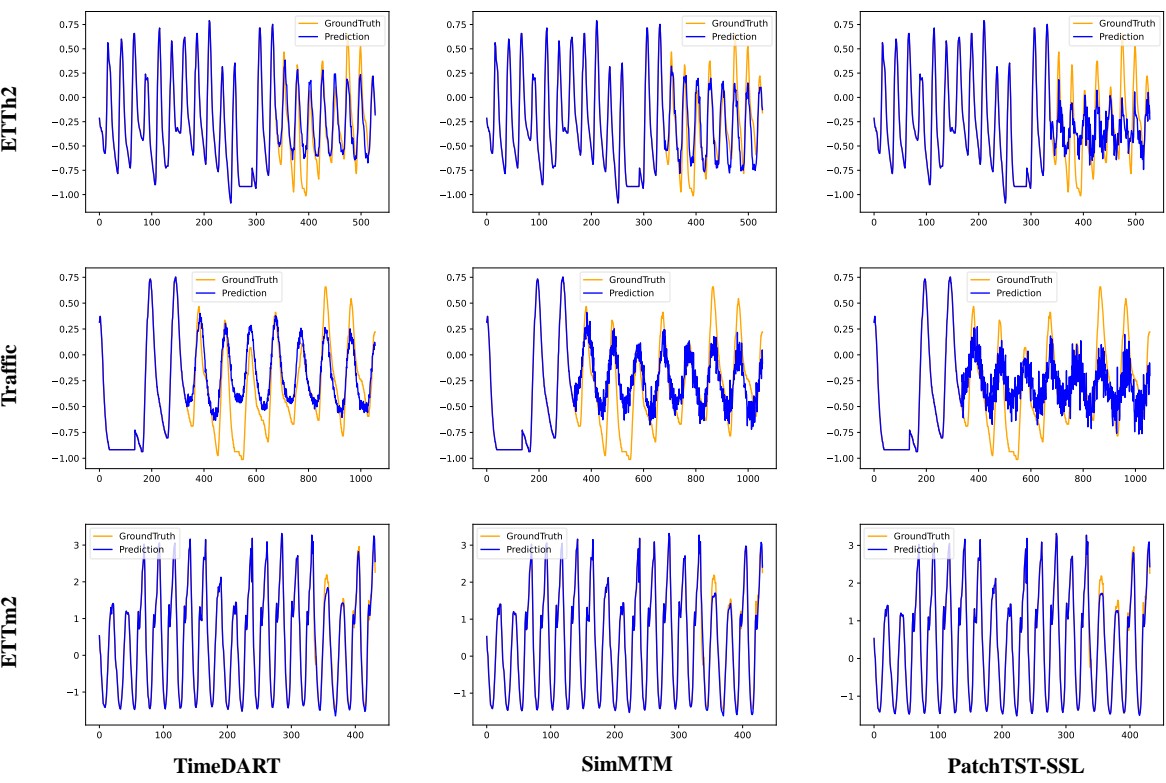

Figure 7: Illustration of forecasting showcases comparing TimeDART and baseline models. The look-back window is set to 336 and the predicted window is set to 192, 96, 720 for the ETTh2, Traffic, and ETTm2 dataset respectively.

In this visualization (Figure 7), TimeDART is compared against SimMTM and PatchTST-SSL, the self-supervised version of PatchTST. The ground truth, input data, and predictions are plotted together. The look-back window is set to 336 for all datasets, while the predicted window varies: 192 for ETTh2, 96 for Traffic, and 720 for ETTm2. This setup ensures that different datasets are forecasted over appropriate future horizons based on their unique characteristics. TimeDART consistently shows more accurate and smoother predictions, closely matching the ground truth compared to the baselines.

## I. Detailed Tables

Table 10: Multivariate time series forecasting full results with look-back window $L = 336$. All results are MSE and MAE from 4 different predicted windows of $\{96, 192, 336, 720\}$. The best results are in **bold** and the second best are underlined. "#1 Counts" represents the number of times the method achieves the best results.

| | | OURS | | | | SELF-SUPERVISED | | | | | | | | SUPERVISED | | | |
|---|---|---|---|---|---|---|---|---|---|---|---|---|---|---|---|---|---|
| METHODS | | **TimeDART** | | RANDOM INIT. | | SimMTM | | PatchTST | | TimeMAE | | CoST | | PatchTST | | DLinear | |
| METRIC | | MSE | MAE | MSE | MAE | MSE | MAE | MSE | MAE | MSE | MAE | MSE | MAE | MSE | MAE | MSE | MAE |
| ETTh1 | 96 | **0.370** | **0.395** | 0.383 | 0.405 | 0.379 | 0.407 | 0.384 | 0.401 | 0.387 | 0.411 | 0.422 | 0.436 | 0.382 | 0.403 | 0.375 | 0.396 |
| | 192 | **0.402** | **0.419** | 0.439 | 0.439 | 0.412 | 0.424 | 0.427 | 0.431 | 0.420 | 0.431 | 0.442 | 0.455 | 0.416 | 0.423 | 0.428 | 0.437 |
| | 336 | 0.426 | **0.427** | 0.467 | 0.457 | 0.421 | 0.431 | 0.461 | 0.450 | 0.453 | 0.453 | 0.472 | 0.462 | 0.441 | 0.440 | 0.448 | 0.449 |
| | 720 | 0.446 | 0.462 | 0.468 | 0.475 | 0.424 | 0.449 | 0.460 | 0.465 | 0.476 | 0.485 | 0.525 | 0.501 | 0.470 | 0.475 | 0.505 | 0.514 |
| ETTh2 | 96 | **0.283** | **0.340** | 0.294 | 0.348 | 0.293 | 0.347 | 0.292 | 0.344 | 0.325 | 0.378 | 0.321 | 0.374 | 0.286 | 0.342 | 0.296 | 0.360 |
| | 192 | **0.343** | **0.381** | 0.357 | 0.390 | 0.355 | 0.386 | 0.354 | 0.391 | 0.394 | 0.423 | 0.380 | 0.403 | 0.357 | 0.389 | 0.391 | 0.423 |
| | 336 | **0.364** | **0.399** | 0.375 | 0.408 | 0.370 | 0.401 | 0.372 | 0.403 | 0.424 | 0.447 | 0.430 | 0.451 | 0.377 | 0.409 | 0.445 | 0.460 |
| | 720 | **0.390** | **0.425** | 0.407 | 0.439 | 0.395 | 0.427 | 0.399 | 0.432 | 0.464 | 0.476 | 0.466 | 0.480 | 0.406 | 0.440 | 0.700 | 0.592 |
| ETTm1 | 96 | **0.286** | **0.342** | 0.301 | 0.354 | 0.288 | 0.348 | 0.289 | 0.344 | 0.289 | 0.344 | 0.291 | 0.343 | 0.298 | 0.345 | 0.303 | 0.346 |
| | 192 | **0.326** | **0.367** | 0.333 | 0.372 | 0.327 | 0.373 | 0.326 | 0.372 | 0.330 | 0.371 | 0.330 | 0.370 | 0.339 | 0.374 | 0.338 | 0.368 |
| | 336 | 0.357 | 0.388 | 0.360 | 0.389 | 0.363 | 0.395 | 0.353 | 0.387 | 0.366 | 0.393 | 0.382 | 0.401 | 0.381 | 0.401 | 0.373 | 0.393 |
| | 720 | 0.407 | **0.417** | 0.408 | 0.418 | 0.412 | 0.424 | 0.399 | 0.418 | 0.416 | 0.424 | 0.422 | 0.425 | 0.428 | 0.431 | 0.428 | 0.423 |
| ETTm2 | 96 | **0.165** | **0.256** | 0.174 | 0.263 | 0.172 | 0.261 | 0.171 | 0.257 | 0.174 | 0.263 | 0.183 | 0.298 | 0.174 | 0.261 | 0.170 | 0.264 |
| | 192 | **0.221** | **0.294** | 0.240 | 0.307 | 0.223 | 0.300 | 0.236 | 0.304 | 0.233 | 0.303 | 0.247 | 0.315 | 0.238 | 0.307 | 0.233 | 0.311 |
| | 336 | **0.279** | **0.330** | 0.284 | 0.334 | 0.282 | 0.331 | 0.291 | 0.344 | 0.291 | 0.340 | 0.298 | 0.346 | 0.293 | 0.346 | 0.298 | 0.358 |
| | 720 | **0.364** | **0.385** | 0.377 | 0.389 | 0.374 | 0.388 | 0.388 | 0.404 | 0.380 | 0.396 | 0.401 | 0.412 | 0.373 | 0.401 | 0.423 | 0.437 |
| ELECTRICITY | 96 | **0.132** | 0.225 | 0.147 | 0.252 | 0.133 | 0.223 | **0.132** | 0.225 | 0.165 | 0.285 | 0.197 | 0.277 | 0.138 | **0.233** | 0.141 | 0.238 |
| | 192 | 0.150 | 0.241 | 0.163 | 0.265 | **0.147** | **0.237** | 0.148 | 0.241 | 0.181 | 0.297 | 0.197 | 0.279 | 0.153 | 0.247 | 0.154 | 0.251 |
| | 336 | **0.166** | **0.258** | 0.179 | 0.280 | 0.166 | 0.265 | 0.167 | 0.260 | 0.199 | 0.312 | 0.211 | 0.295 | 0.170 | 0.263 | 0.170 | 0.269 |
| | 720 | **0.203** | **0.290** | 0.218 | 0.312 | 0.203 | 0.297 | 0.205 | 0.292 | 0.238 | 0.341 | 0.255 | 0.330 | 0.206 | 0.295 | 0.205 | 0.302 |
| TRAFFIC | 96 | **0.357** | **0.247** | 0.386 | 0.267 | 0.368 | 0.262 | 0.382 | 0.262 | 0.382 | 0.261 | 0.378 | 0.365 | 0.395 | 0.272 | 0.411 | 0.284 |
| | 192 | 0.376 | **0.256** | 0.398 | 0.269 | 0.373 | **0.251** | 0.385 | 0.261 | 0.399 | 0.267 | **0.371** | 0.352 | 0.411 | 0.278 | 0.423 | 0.289 |
| | 336 | **0.389** | **0.262** | 0.410 | 0.274 | 0.395 | **0.254** | 0.409 | 0.275 | 0.411 | 0.274 | 0.467 | 0.354 | 0.424 | 0.284 | 0.437 | 0.297 |
| | 720 | **0.429** | **0.286** | 0.446 | 0.299 | 0.432 | 0.290 | 0.438 | 0.291 | 0.446 | 0.298 | 0.525 | 0.378 | 0.453 | 0.300 | 0.467 | 0.316 |
| WEATHER | 96 | 0.149 | 0.199 | 0.155 | 0.206 | 0.158 | 0.211 | 0.148 | **0.196** | 0.154 | 0.203 | 0.162 | 0.235 | **0.147** | 0.197 | 0.176 | 0.236 |
| | 192 | 0.193 | **0.240** | 0.198 | 0.246 | 0.199 | 0.249 | 0.193 | **0.240** | 0.191 | 0.241 | 0.212 | 0.255 | **0.191** | **0.240** | 0.217 | 0.275 |
| | 336 | 0.244 | 0.280 | 0.250 | 0.286 | 0.246 | 0.286 | 0.244 | **0.279** | 0.243 | 0.282 | 0.258 | 0.293 | 0.244 | 0.282 | 0.264 | 0.315 |
| | 720 | **0.317** | **0.331** | 0.319 | 0.335 | 0.317 | 0.337 | 0.321 | 0.334 | 0.318 | 0.334 | 0.334 | 0.346 | 0.320 | 0.334 | 0.325 | 0.364 |
| EXCHANGE | 96 | **0.086** | 0.211 | 0.102 | 0.229 | 0.100 | 0.226 | 0.088 | **0.207** | 0.098 | 0.226 | 0.102 | 0.229 | 0.094 | 0.213 | 0.087 | 0.217 |
| | 192 | 0.175 | 0.302 | 0.224 | 0.343 | 0.210 | 0.332 | 0.186 | 0.308 | 0.219 | 0.340 | 0.212 | 0.334 | 0.191 | 0.311 | **0.164** | **0.298** |
| | 336 | 0.344 | 0.431 | 0.384 | 0.453 | 0.389 | 0.460 | 0.374 | 0.446 | 0.400 | 0.466 | 0.384 | 0.452 | 0.343 | **0.427** | 0.333 | 0.437 |
| | 720 | **0.829** | **0.675** | 1.051 | 0.774 | 1.104 | 0.800 | 0.857 | 0.692 | 0.989 | 0.751 | 1.124 | 0.805 | 0.888 | 0.706 | 0.988 | 0.749 |
| #1 COUNTS | | 43 | | 0 | | 10 | | 9 | | 2 | | 1 | | 5 | | 3 | |

Table 11: Multivariate time series forecasting full results on PEMS datasets. All results are MSE and MAE from 4 different predicted windows of $\{12, 24, 36, 48\}$. The best results are in **bold** and the second best are underlined. * means we doesn't apply instance normalization on both pre-training and fine-tuning.

| | Methods | OURS | | | | SELF-SUPERVISED | | | | | | | | SUPERVISED | | | |
| | | **TimeDART** | | Random Init. | | SimMTM | | PatchTST | | TimeMAE | | CoST | | PatchTST | | DLinear | |
| | Metrics | MSE | MAE | MSE | MAE | MSE | MAE | MSE | MAE | MSE | MAE | MSE | MAE | MSE | MAE | MSE | MAE |
|---|---|---|---|---|---|---|---|---|---|---|---|---|---|---|---|---|---|
| PEMS03 | 12 | 0.082 | **0.188** | 0.089 | 0.198 | 0.082 | 0.189 | **0.081** | 0.191 | 0.093 | 0.207 | 0.102 | 0.213 | 0.099 | 0.212 | 0.120 | 0.240 |
| | 24 | **0.127** | **0.236** | 0.138 | 0.248 | 0.133 | 0.239 | 0.131 | 0.239 | 0.138 | 0.249 | 0.142 | 0.251 | 0.139 | 0.251 | 0.198 | 0.312 |
| | 36 | **0.176** | **0.282** | 0.187 | 0.292 | 0.183 | 0.286 | 0.181 | 0.287 | 0.186 | 0.292 | 0.189 | 0.298 | 0.211 | 0.320 | 0.330 | 0.420 |
| | 48 | **0.223** | **0.321** | 0.241 | 0.326 | 0.233 | 0.326 | 0.231 | 0.325 | 0.242 | 0.327 | 0.244 | 0.331 | 0.262 | 0.370 | 0.460 | 0.520 |
| PEMS04 | 12* | **0.087** | **0.197** | 0.097 | 0.207 | 0.100 | 0.205 | 0.091 | 0.202 | 0.098 | 0.205 | 0.097 | 0.207 | 0.101 | 0.212 | 0.145 | 0.270 |
| | 24* | **0.121** | **0.235** | 0.128 | 0.243 | 0.125 | 0.241 | 0.126 | 0.240 | 0.126 | 0.242 | 0.133 | 0.251 | 0.135 | 0.265 | 0.220 | 0.335 |
| | 36* | **0.149** | **0.260** | 0.161 | 0.275 | 0.156 | 0.271 | 0.157 | 0.266 | 0.160 | 0.274 | 0.167 | 0.278 | 0.162 | 0.276 | 0.350 | 0.430 |
| | 48* | **0.176** | **0.287** | 0.192 | 0.296 | 0.190 | 0.293 | 0.183 | 0.289 | 0.193 | 0.301 | 0.192 | 0.311 | 0.199 | 0.312 | 0.446 | 0.489 |
| PEMS07 | 12* | **0.066** | **0.174** | 0.078 | 0.183 | 0.072 | 0.180 | 0.071 | 0.178 | 0.074 | 0.181 | 0.076 | 0.186 | 0.076 | 0.185 | 0.109 | 0.238 |
| | 24* | **0.109** | **0.218** | 0.117 | 0.226 | 0.111 | 0.220 | 0.115 | 0.224 | 0.116 | 0.225 | 0.118 | 0.229 | 0.129 | 0.242 | 0.206 | 0.322 |
| | 36* | **0.147** | **0.250** | 0.156 | 0.261 | 0.151 | 0.254 | 0.149 | 0.252 | 0.155 | 0.259 | 0.158 | 0.263 | 0.178 | 0.272 | 0.395 | 0.453 |
| | 48* | **0.188** | **0.286** | 0.200 | 0.301 | 0.191 | 0.291 | 0.193 | 0.294 | 0.202 | 0.298 | 0.203 | 0.300 | 0.211 | 0.314 | 0.578 | 0.534 |
| PEMS08 | 12 | **0.105** | **0.219** | 0.115 | 0.226 | 0.110 | 0.223 | 0.110 | 0.225 | 0.112 | 0.226 | 0.112 | 0.224 | 0.125 | 0.232 | 0.148 | 0.271 |
| | 24 | **0.179** | **0.286** | 0.189 | 0.298 | 0.182 | 0.289 | 0.183 | 0.290 | 0.184 | 0.291 | 0.192 | 0.299 | 0.204 | 0.291 | 0.239 | 0.345 |
| | 36* | **0.239** | **0.296** | 0.251 | 0.310 | 0.246 | 0.300 | 0.244 | 0.299 | 0.248 | 0.309 | 0.252 | 0.315 | 0.275 | 0.314 | 0.422 | 0.454 |
| | 48* | **0.281** | **0.328** | 0.295 | 0.338 | 0.287 | 0.333 | 0.288 | 0.333 | 0.299 | 0.341 | 0.303 | 0.342 | 0.316 | 0.343 | 0.628 | 0.539 |

Table 12: Multivariate time series full results of general pre-training and fine-tuning across different domains. All results are MSE and MAE from 4 different predicted windows of $\{12, 24, 36, 48\}$ for PEMS datasets and $\{96, 192, 336, 720\}$ for others. The best results are in **bold** and the second best are underlined.

| Methods Domain | Metric | | **TimeDART** | | Random Init. | | SimMTM | | PatchTST | | TimeMAE | | CoST | |
| | | | MSE | MAE | MSE | MAE | MSE | MAE | MSE | MAE | MSE | MAE | MSE | MAE |
|---|---|---|---|---|---|---|---|---|---|---|---|---|---|---|---|
| Energy | ETTm2 | 96 | **0.164** | **0.255** | 0.175 | 0.269 | 0.169 | 0.260 | 0.171 | 0.264 | 0.175 | 0.272 | 0.182 | 0.289 |
| | | 192 | **0.219** | **0.291** | 0.241 | 0.313 | 0.232 | 0.303 | 0.238 | 0.311 | 0.239 | 0.312 | 0.254 | 0.319 |
| | | 336 | **0.274** | **0.325** | 0.283 | 0.336 | 0.276 | 0.332 | 0.280 | 0.334 | 0.284 | 0.339 | 0.293 | 0.342 |
| | | 720 | **0.361** | **0.381** | 0.374 | 0.392 | 0.369 | 0.387 | 0.371 | 0.389 | 0.376 | 0.394 | 0.382 | 0.397 |
| Weather | Weather | 96 | 0.150 | 0.201 | 0.161 | 0.211 | 0.159 | 0.209 | 0.153 | **0.199** | **0.149** | **0.199** | 0.172 | 0.219 |
| | | 192 | **0.195** | 0.239 | 0.212 | 0.247 | 0.202 | **0.238** | 0.199 | 0.239 | 0.196 | 0.239 | 0.243 | 0.258 |
| | | 336 | 0.246 | **0.278** | 0.253 | 0.290 | 0.249 | 0.285 | **0.245** | 0.280 | 0.249 | 0.281 | 0.267 | 0.302 |
| | | 720 | **0.318** | **0.333** | 0.325 | 0.341 | 0.322 | 0.338 | 0.319 | 0.335 | 0.320 | 0.334 | 0.333 | 0.358 |
| Finance | Exchange | 96 | **0.078** | **0.192** | 0.113 | 0.234 | 0.111 | 0.234 | 0.084 | 0.202 | 0.094 | 0.211 | 0.113 | 0.239 |
| | | 192 | **0.168** | **0.296** | 0.235 | 0.367 | 0.232 | 0.364 | 0.179 | 0.289 | 0.182 | 0.292 | 0.243 | 0.372 |
| | | 336 | **0.355** | **0.442** | 0.402 | 0.467 | 0.392 | 0.455 | 0.362 | 0.451 | 0.378 | 0.459 | 0.413 | 0.472 |
| | | 720 | **0.798** | **0.667** | 1.078 | 0.823 | 1.023 | 0.812 | 0.889 | 0.701 | 0.972 | 0.734 | 1.143 | 0.854 |
| Transportation | PEMS04 | 96 | **0.079** | **0.185** | 0.095 | 0.199 | 0.089 | 0.192 | 0.087 | 0.192 | 0.092 | 0.193 | 0.094 | 0.196 |
| | | 192 | **0.116** | **0.226** | 0.124 | 0.239 | 0.121 | 0.235 | 0.120 | 0.234 | 0.122 | 0.234 | 0.126 | 0.239 |
| | | 336 | **0.142** | **0.254** | 0.158 | 0.273 | 0.149 | 0.262 | 0.151 | 0.259 | 0.157 | 0.273 | 0.162 | 0.275 |
| | | 720 | **0.169** | **0.279** | 0.185 | 0.291 | 0.173 | 0.285 | 0.175 | 0.288 | 0.182 | 0.290 | 0.187 | 0.296 |

Table 13: Full result of TCN as backbone.

| Methods Backbone | Metrics | ETTh2 | | | | ETTm2 | | | | Electricity | | | | PEMS04 | | | |
| | | 96 | 192 | 336 | 720 | 96 | 192 | 336 | 720 | 96 | 192 | 336 | 720 | 12 | 24 | 36 | 48 |
|---|---|---|---|---|---|---|---|---|---|---|---|---|---|---|---|---|---|
| TCN | MSE | 0.285 | 0.342 | 0.368 | 0.400 | 0.172 | 0.226 | 0.282 | 0.372 | 0.134 | 0.153 | 0.166 | 0.205 | 0.086 | 0.119 | 0.152 | 0.179 |
| | MAE | 0.346 | 0.388 | 0.413 | 0.435 | 0.265 | 0.299 | 0.337 | 0.390 | 0.223 | 0.241 | 0.259 | 0.291 | 0.195 | 0.235 | 0.264 | 0.289 |
| Random Init. | MSE | 0.287 | 0.352 | 0.380 | 0.407 | 0.176 | 0.238 | 0.284 | 0.378 | 0.143 | 0.165 | 0.180 | 0.221 | 0.098 | 0.126 | 0.163 | 0.194 |
| | MAE | 0.349 | 0.393 | 0.423 | 0.445 | 0.272 | 0.304 | 0.335 | 0.392 | 0.249 | 0.266 | 0.281 | 0.315 | 0.206 | 0.243 | 0.278 | 0.297 |
| Transformer | MSE | 0.283 | 0.345 | 0.365 | 0.390 | 0.165 | 0.221 | 0.279 | 0.364 | 0.132 | 0.150 | 0.166 | 0.203 | 0.087 | 0.121 | 0.149 | 0.176 |
| | MAE | 0.340 | 0.382 | 0.399 | 0.425 | 0.256 | 0.294 | 0.330 | 0.385 | 0.225 | 0.241 | 0.258 | 0.290 | 0.197 | 0.235 | 0.260 | 0.287 |

Table 14: Full result of the ablation study on datasets for forecasting. The best results are in **bold**.

| METHODS / METRICS | | TIMEDART MSE | MAE | w/o AR MSE | MAE | w/o DIFF MSE | MAE | w/o AR-DIFF MSE | MAE | RANDOM INIT. MSE | MAE |
|---|---|---|---|---|---|---|---|---|---|---|---|
| ETTH2 | 96 | **0.283** | **0.340** | 0.299 | 0.352 | 0.288 | 0.343 | 0.300 | 0.354 | 0.294 | 0.348 |
| | 192 | **0.345** | **0.382** | 0.364 | 0.390 | 0.351 | 0.384 | 0.365 | 0.390 | 0.357 | 0.390 |
| | 336 | **0.365** | **0.399** | 0.387 | 0.414 | 0.372 | 0.404 | 0.386 | 0.413 | 0.375 | 0.390 |
| | 720 | **0.390** | **0.425** | 0.409 | 0.438 | 0.396 | 0.432 | 0.406 | 0.436 | 0.407 | 0.439 |
| AVG. | | **0.346** | **0.387** | 0.365 | 0.399 | 0.352 | 0.391 | 0.364 | 0.398 | 0.358 | 0.396 |
| ETTM2 | 96 | **0.165** | **0.256** | 0.184 | 0.276 | 0.175 | 0.265 | 0.186 | 0.278 | 0.174 | 0.263 |
| | 192 | **0.221** | **0.294** | 0.245 | 0.317 | 0.228 | 0.300 | 0.246 | 0.318 | 0.240 | 0.307 |
| | 336 | **0.279** | **0.330** | 0.307 | 0.355 | 0.281 | 0.331 | 0.311 | 0.367 | 0.284 | 0.334 |
| | 720 | **0.364** | **0.385** | 0.388 | 0.403 | 0.374 | 0.392 | 0.395 | 0.420 | 0.377 | 0.389 |
| AVG. | | **0.257** | **0.316** | 0.271 | 0.326 | 0.265 | 0.322 | 0.285 | 0.346 | 0.269 | 0.323 |
| ELECTRICITY | 96 | **0.132** | **0.225** | 0.163 | 0.281 | 0.134 | 0.228 | 0.158 | 0.276 | 0.147 | 0.252 |
| | 192 | **0.150** | **0.241** | 0.179 | 0.294 | **0.150** | 0.242 | 0.163 | 0.265 | 0.163 | 0.265 |
| | 336 | **0.166** | **0.258** | 0.195 | 0.306 | 0.167 | 0.259 | 0.199 | 0.312 | 0.179 | 0.280 |
| | 720 | **0.203** | **0.290** | 0.234 | 0.335 | 0.205 | 0.292 | 0.238 | 0.341 | 0.218 | 0.312 |
| AVG. | | **0.163** | **0.254** | 0.181 | 0.279 | 0.168 | 0.258 | 0.190 | 0.299 | 0.177 | 0.277 |
| PEMS04 | 12 | **0.087** | **0.197** | 0.102 | 0.214 | 0.097 | 0.208 | 0.103 | 0.215 | 0.097 | 0.207 |
| | 24 | **0.121** | **0.235** | 0.130 | 0.245 | 0.129 | 0.244 | 0.132 | 0.246 | 0.128 | 0.243 |
| | 36 | **0.149** | **0.260** | 0.156 | 0.269 | 0.160 | 0.275 | 0.164 | 0.278 | 0.161 | 0.275 |
| | 48 | **0.176** | **0.287** | 0.189 | 0.292 | 0.193 | 0.298 | 0.195 | 0.300 | 0.192 | 0.296 |
| AVG. | | **0.133** | **0.245** | 0.144 | 0.255 | 0.145 | 0.256 | 0.149 | 0.260 | 0.145 | 0.255 |

Table 15: Full result of the mask ratio on datasets for forecasting. The best results are in **bold**.

| MASK RATIO / METRICS | | 1.0 MSE | MAE | 0.7 MSE | MAE | 0.5 MSE | MAE | 0.0 MSE | MAE | RANDOM INIT. MSE | MAE |
|---|---|---|---|---|---|---|---|---|---|---|---|
| ETTH2 | 96 | **0.283** | **0.340** | 0.285 | 0.342 | 0.289 | 0.346 | 0.294 | 0.347 | 0.294 | 0.348 |
| | 192 | **0.345** | **0.382** | 0.348 | 0.384 | 0.348 | 0.384 | 0.356 | 0.389 | 0.357 | 0.390 |
| | 336 | 0.365 | 0.399 | **0.364** | **0.398** | 0.368 | 0.402 | 0.372 | 0.405 | 0.375 | 0.390 |
| | 720 | **0.390** | **0.425** | 0.395 | 0.428 | 0.398 | 0.433 | 0.400 | 0.434 | 0.407 | 0.439 |
| AVG. | | **0.346** | **0.387** | 0.348 | 0.388 | 0.351 | 0.391 | 0.356 | 0.394 | 0.358 | 0.396 |
| ETTM2 | 96 | **0.165** | **0.256** | 0.167 | 0.258 | **0.165** | 0.257 | 0.172 | 0.259 | 0.174 | 0.263 |
| | 192 | **0.221** | **0.294** | 0.224 | 0.298 | 0.230 | 0.302 | 0.235 | 0.304 | 0.240 | 0.307 |
| | 336 | **0.279** | **0.330** | 0.281 | 0.334 | 0.281 | 0.335 | 0.282 | 0.335 | 0.284 | 0.334 |
| | 720 | **0.364** | **0.385** | **0.364** | **0.385** | 0.368 | 0.390 | 0.370 | 0.392 | 0.377 | 0.389 |
| AVG. | | **0.257** | **0.316** | 0.259 | 0.319 | 0.261 | 0.321 | 0.265 | 0.323 | 0.269 | 0.323 |
| TRAFFIC | 96 | **0.357** | **0.247** | **0.357** | **0.247** | 0.360 | 0.250 | 0.365 | 0.256 | 0.386 | 0.267 |
| | 192 | **0.376** | **0.256** | 0.378 | 0.259 | 0.381 | 0.263 | 0.385 | 0.267 | 0.398 | 0.269 |
| | 336 | **0.389** | **0.262** | 0.390 | 0.264 | 0.392 | 0.267 | 0.398 | 0.272 | 0.410 | 0.274 |
| | 720 | **0.429** | **0.286** | 0.432 | 0.289 | 0.435 | 0.293 | 0.441 | 0.298 | 0.446 | 0.299 |
| AVG. | | **0.388** | **0.263** | 0.389 | 0.265 | 0.392 | 0.268 | 0.397 | 0.273 | 0.410 | 0.277 |
| PEMS04 | 12 | **0.087** | **0.197** | **0.087** | **0.197** | 0.090 | 0.199 | 0.092 | 0.202 | 0.097 | 0.207 |
| | 24 | **0.121** | **0.235** | 0.122 | 0.236 | 0.125 | 0.240 | 0.128 | 0.242 | 0.128 | 0.243 |
| | 36 | **0.149** | **0.260** | 0.151 | 0.262 | 0.153 | 0.264 | 0.156 | 0.266 | 0.161 | 0.275 |
| | 48 | **0.176** | **0.287** | **0.176** | 0.288 | 0.177 | **0.287** | 0.178 | 0.291 | 0.192 | 0.296 |
| AVG. | | **0.133** | **0.245** | 0.134 | 0.246 | 0.136 | 0.248 | 0.139 | 0.250 | 0.145 | 0.255 |

Table 16: Full result of hyperparameter analysis of noise steps and noise schedulers. The best results are in **bold**.

| PARAM. METRIC | | TOTAL NOISE STEPS | | | | | | NOISE SCHEDULER | | | | RANDOM INIT. | |
|---|---|---|---|---|---|---|---|---|---|---|---|---|---|
| | | 750 | | 1000 | | 1250 | | Cos. | | Lin. | | | |
| | | MSE | MAE | MSE | MAE | MSE | MAE | MSE | MAE | MSE | MAE | MSE | MAE |
| ETTM2 | 96 | 0.167 | 0.259 | **0.165** | **0.256** | 0.164 | 0.255 | **0.165** | **0.256** | 0.171 | 0.260 | 0.174 | 0.263 |
| | 192 | 0.222 | 0.293 | **0.221** | **0.294** | 0.220 | 0.292 | **0.221** | **0.294** | 0.238 | 0.305 | 0.240 | 0.307 |
| | 336 | 0.280 | 0.331 | **0.279** | **0.330** | 0.279 | 0.329 | **0.279** | **0.330** | 0.281 | 0.333 | 0.284 | 0.334 |
| | 720 | **0.364** | 0.386 | **0.364** | **0.385** | 0.364 | 0.385 | 0.364 | 0.385 | 0.372 | 0.383 | 0.377 | 0.389 |
| PEMS04 | 12 | 0.089 | 0.198 | **0.087** | **0.197** | 0.088 | 0.198 | **0.087** | **0.197** | 0.096 | 0.207 | 0.097 | 0.207 |
| | 24 | **0.120** | **0.235** | 0.121 | **0.235** | 0.122 | 0.236 | **0.121** | **0.235** | 0.126 | 0.242 | 0.128 | 0.243 |
| | 36 | 0.150 | 0.261 | **0.149** | **0.260** | 0.151 | 0.263 | **0.149** | **0.260** | 0.159 | 0.275 | 0.161 | 0.275 |
| | 48 | 0.177 | 0.287 | **0.176** | **0.287** | 0.177 | 0.289 | **0.176** | **0.287** | 0.189 | 0.293 | 0.192 | 0.296 |

Table 17: Full result of hyperparameter analysis of the layer numbers of denoising decoder. The best results are in **bold**.

| NUMBERS | | 0 | | 1 | | 2 | | 3 | | RANDOM INIT. | |
|---|---|---|---|---|---|---|---|---|---|---|---|
| METRIC | | MSE | MAE | MSE | MAE | MSE | MAE | MSE | MAE | MSE | MAE |
| ETTH2 | 96 | 0.288 | 0.343 | **0.283** | **0.340** | 0.284 | 0.345 | 0.284 | 0.345 | 0.294 | 0.348 |
| | 192 | 0.351 | 0.384 | 0.343 | **0.381** | **0.342** | 0.382 | **0.342** | 0.382 | 0.357 | 0.390 |
| | 336 | 0.372 | 0.404 | 0.364 | 0.399 | **0.360** | **0.398** | 0.361 | 0.400 | 0.375 | 0.408 |
| | 720 | 0.396 | 0.432 | **0.390** | **0.425** | 0.394 | 0.428 | 0.397 | 0.433 | 0.407 | 0.439 |
| AVG. | | 0.352 | 0.391 | **0.345** | **0.386** | 0.345 | 0.388 | 0.346 | 0.390 | 0.358 | 0.396 |
| ETTM2 | 96 | 0.175 | 0.265 | **0.165** | **0.256** | 0.166 | 0.257 | 0.167 | 0.257 | 0.174 | 0.263 |
| | 192 | 0.228 | 0.300 | **0.221** | **0.294** | 0.226 | 0.297 | 0.230 | 0.399 | 0.240 | 0.307 |
| | 336 | 0.281 | 0.331 | **0.279** | **0.330** | 0.280 | 0.333 | 0.282 | 0.337 | 0.284 | 0.334 |
| | 720 | 0.374 | 0.392 | **0.364** | **0.385** | 0.372 | 0.388 | 0.379 | 0.394 | 0.377 | 0.389 |
| AVG. | | 0.265 | 0.322 | **0.257** | **0.316** | 0.261 | 0.319 | 0.265 | 0.347 | 0.269 | 0.323 |
| ELECTRICITY | 96 | 0.136 | 0.229 | **0.132** | **0.225** | 0.134 | 0.227 | 0.142 | 0.244 | 0.147 | 0.252 |
| | 192 | 0.154 | 0.245 | **0.150** | **0.241** | 0.151 | 0.243 | 0.160 | 0.260 | 0.163 | 0.265 |
| | 336 | 0.170 | 0.263 | **0.166** | **0.258** | 0.169 | 0.258 | 0.175 | 0.274 | 0.179 | 0.280 |
| | 720 | 0.212 | 0.296 | **0.203** | **0.290** | 0.211 | 0.304 | 0.215 | 0.310 | 0.218 | 0.312 |
| AVG. | | 0.168 | 0.258 | **0.163** | **0.254** | 0.166 | 0.258 | 0.173 | 0.272 | 0.177 | 0.277 |
| PEMS04 | 12 | 0.097 | 0.208 | 0.087 | 0.197 | **0.086** | **0.196** | 0.091 | 0.201 | 0.097 | 0.207 |
| | 24 | 0.129 | 0.244 | **0.121** | **0.235** | 0.121 | 0.235 | 0.125 | 0.241 | 0.128 | 0.243 |
| | 36 | 0.160 | 0.275 | 0.149 | **0.260** | **0.148** | 0.261 | 0.153 | 0.265 | 0.161 | 0.275 |
| | 48 | 0.193 | 0.298 | **0.176** | **0.287** | 0.175 | 0.287 | 0.181 | 0.293 | 0.192 | 0.296 |
| AVG. | | 0.145 | 0.256 | **0.133** | **0.245** | 0.133 | 0.245 | 0.138 | 0.250 | 0.145 | 0.255 |
| METRIC | | ACC. | F1 | ACC. | F1 | ACC. | F1 | ACC. | F1 | ACC. | F1 |
| HAR | - | 0.9002 | 0.9028 | **0.9247** | **0.9286** | 0.9186 | 0.9203 | 0.9048 | 0.9059 | 0.8931 | 0.8946 |

Table 18: Full result of hyperparameter sensitivity analysis of noise addition process and whether clean patches and noise-added patches share the same embedding layer. The best results are in **bold**.

| Param. / Metric | | Noise Addition Process | | | | Whether to Share Embedding Layer | | | | Random Init. | |
| --- | --- | --- | --- | --- | --- | --- | --- | --- | --- | --- | --- |
| | | Independent Steps | | Same Step | | Shared Embedding | | Separate Embedding | | | |
| | | MSE | MAE | MSE | MAE | MSE | MAE | MSE | MAE | MSE | MAE |
| ETTm2 | 96 | **0.165** | **0.256** | 0.168 | 0.259 | **0.165** | **0.256** | 0.170 | 0.258 | 0.174 | 0.263 |
| | 192 | **0.221** | **0.294** | 0.223 | 0.295 | **0.221** | **0.294** | 0.223 | 0.297 | 0.240 | 0.307 |
| | 336 | **0.279** | **0.330** | 0.281 | 0.334 | **0.279** | **0.330** | 0.284 | 0.334 | 0.284 | 0.334 |
| | 720 | **0.364** | **0.385** | **0.364** | 0.386 | **0.364** | **0.385** | 0.368 | 0.389 | 0.377 | 0.389 |
| Avg. | | **0.257** | **0.316** | 0.259 | 0.319 | **0.257** | **0.316** | 0.261 | 0.320 | 0.269 | 0.323 |
| PEMS04 | 12 | **0.087** | **0.197** | 0.088 | 0.199 | **0.087** | **0.197** | **0.087** | **0.197** | 0.097 | 0.207 |
| | 24 | **0.121** | **0.235** | 0.123 | 0.237 | **0.121** | **0.235** | 0.122 | 0.237 | 0.128 | 0.243 |
| | 36 | **0.149** | **0.260** | 0.154 | 0.263 | **0.149** | **0.260** | 0.152 | 0.263 | 0.161 | 0.275 |
| | 48 | **0.176** | **0.287** | 0.180 | 0.290 | **0.176** | **0.287** | 0.178 | 0.289 | 0.192 | 0.296 |
| Avg. | | **0.133** | **0.245** | 0.136 | 0.247 | **0.133** | **0.245** | 0.135 | 0.247 | 0.145 | 0.255 |
| HAR | | **0.9247** | **0.9286** | 0.9168 | 0.9112 | **0.9247** | **0.9286** | 0.9104 | 0.8993 | 0.8931 | 0.8946 |

Table 19: Full result of hyperparameter sensitivity analysis of patch length. The best results are in **bold**.

| Length | | 1 | | 2 | | 4 | | 8 | | 16 | | Random Init. | |
| --- | --- | --- | --- | --- | --- | --- | --- | --- | --- | --- | --- | --- | --- |
| Metric | | MSE | MAE | MSE | MAE | MSE | MAE | MSE | MAE | MSE | MAE | MSE | MAE |
| ETTh2 | 96 | 0.312 | 0.364 | **0.283** | **0.340** | 0.295 | 0.348 | 0.301 | 0.356 | 0.313 | 0.365 | 0.294 | 0.348 |
| | 192 | 0.387 | 0.412 | **0.343** | **0.381** | 0.348 | 0.385 | 0.356 | 0.390 | 0.365 | 0.400 | 0.357 | 0.390 |
| | 336 | 0.419 | 0.439 | **0.364** | **0.399** | 0.369 | 0.406 | 0.370 | 0.407 | 0.377 | 0.415 | 0.375 | 0.390 |
| | 720 | 0.452 | 0.469 | **0.390** | **0.425** | 0.399 | 0.434 | 0.403 | 0.436 | 0.412 | 0.443 | 0.407 | 0.439 |
| Avg. | | 0.393 | 0.421 | **0.345** | **0.386** | 0.353 | 0.393 | 0.358 | 0.397 | 0.367 | 0.406 | 0.358 | 0.396 |
| ETTm2 | 96 | 0.169 | 0.258 | **0.165** | **0.256** | 0.177 | 0.267 | 0.168 | 0.258 | 0.170 | 0.261 | 0.174 | 0.263 |
| | 192 | 0.226 | 0.295 | **0.221** | **0.294** | 0.231 | 0.302 | 0.226 | 0.297 | 0.224 | 0.297 | 0.240 | 0.307 |
| | 336 | 0.283 | 0.333 | 0.279 | **0.330** | 0.284 | 0.336 | 0.278 | **0.330** | **0.277** | **0.330** | 0.284 | 0.334 |
| | 720 | 0.371 | 0.388 | 0.364 | 0.385 | 0.378 | 0.392 | **0.362** | **0.382** | 0.370 | 0.385 | 0.377 | 0.389 |
| Avg. | | 0.262 | 0.3185 | **0.257** | **0.316** | 0.259 | 0.317 | 0.260 | 0.318 | 0.268 | 0.324 | 0.269 | 0.323 |
| Electricity | 96 | 0.165 | 0.285 | 0.149 | 0.254 | 0.135 | 0.234 | **0.132** | **0.225** | 0.146 | 0.250 | 0.147 | 0.252 |
| | 192 | 0.181 | 0.297 | 0.163 | 0.266 | 0.152 | 0.249 | **0.150** | **0.241** | 0.161 | 0.264 | 0.240 | 0.307 |
| | 336 | 0.199 | 0.312 | 0.180 | 0.282 | 0.169 | 0.266 | **0.166** | **0.258** | 0.178 | 0.281 | 0.284 | 0.334 |
| | 720 | 0.238 | 0.341 | 0.220 | 0.313 | 0.208 | 0.299 | **0.203** | **0.290** | 0.218 | 0.313 | 0.377 | 0.389 |
| Avg. | | 0.196 | 0.309 | 0.178 | 0.279 | 0.166 | 0.262 | **0.163** | **0.254** | 0.176 | 0.277 | 0.177 | 0.277 |
| PEMS04 | 12 | 0.088 | 0.198 | **0.087** | **0.197** | 0.089 | 0.198 | 0.096 | 0.207 | 0.099 | 0.209 | 0.097 | 0.207 |
| | 24 | 0.122 | 0.236 | 0.121 | **0.235** | **0.120** | **0.235** | 0.126 | 0.242 | 0.131 | 0.252 | 0.128 | 0.243 |
| | 36 | 0.151 | 0.263 | **0.149** | **0.260** | 0.150 | 0.261 | 0.159 | 0.275 | 0.165 | 0.276 | 0.161 | 0.275 |
| | 48 | 0.177 | 0.289 | **0.176** | **0.287** | 0.177 | **0.287** | 0.189 | 0.293 | 0.197 | 0.303 | 0.192 | 0.296 |
| Avg. | | 0.135 | 0.247 | **0.133** | **0.245** | 0.134 | **0.245** | 0.143 | 0.254 | 0.148 | 0.260 | 0.145 | 0.255 |
| Length | | 2 | | 4 | | 8 | | 16 | | 32 | | Random Init. | |
| Metric | | Acc. | F1 | Acc. | F1 | Acc. | F1 | Acc. | F1 | Acc. | F1 | Acc. | F1 |
| HAR | - | 0.8969 | 0.8952 | 0.9132 | 0.9145 | **0.9247** | **0.9286** | 0.9157 | 0.9168 | 0.9023 | 0.9012 | 0.8931 | 0.8946 |

Table 20: Full result of few-shot fine-tuning.

| Portion | | 5% | | 10% | | 100% | | Random Init. | |
|---|---|---|---|---|---|---|---|---|---|
| Metric | | MSE | MAE | MSE | MAE | MSE | MAE | MSE | MAE |
| ETTH2 | 96 | 0.292 | 0.346 | 0.289 | 0.344 | **0.283** | **0.340** | 0.294 | 0.348 |
| | 192 | 0.356 | 0.389 | 0.354 | 0.387 | **0.345** | **0.382** | 0.357 | 0.390 |
| | 336 | 0.375 | 0.405 | 0.374 | 0.403 | **0.365** | **0.399** | 0.375 | 0.408 |
| | 720 | 0.399 | 0.429 | 0.397 | 0.428 | **0.390** | **0.425** | 0.407 | 0.439 |
| | Avg. | 0.356 | 0.392 | 0.354 | 0.391 | **0.346** | **0.387** | 0.358 | 0.396 |
| PEMS04 | 12 | 0.093 | 0.205 | 0.091 | 0.205 | **0.087** | **0.197** | 0.097 | 0.207 |
| | 24 | 0.128 | 0.244 | 0.127 | 0.242 | **0.121** | **0.235** | 0.128 | 0.243 |
| | 36 | 0.158 | 0.272 | 0.157 | 0.269 | **0.149** | **0.260** | 0.161 | 0.275 |
| | 48 | 0.187 | 0.294 | 0.185 | 0.285 | **0.176** | **0.287** | 0.192 | 0.296 |
| | Avg. | 0.142 | 0.254 | 0.140 | 0.250 | **0.133** | **0.245** | 0.145 | 0.255 |
| Portion | | 5% | | 10% | | 100% | | Random Init. | |
| Metric | | Acc. | F1 Score | Acc. | F1 Score | Acc. | F1 Score | Acc. | F1 Score |
| HAR | | 0.8843 | 0.8901 | 0.9014 | 0.9053 | **0.9247** | **0.9286** | 0.8738 | 0.8723 |

Table 21: Full result of linear probing fine-tuning.

| Linear Probing | | Random Init. | | TimeDART | | SimMTM | | TimeMAE | |
|---|---|---|---|---|---|---|---|---|---|
| Metrics | | MSE | MAE | MSE | MAE | MSE | MAE | MSE | MAE |
| ETTH2 | 96 | 0.297 | 0.345 | **0.281** | **0.338** | **0.281** | 0.339 | 0.289 | 0.342 |
| | 192 | 0.368 | 0.397 | **0.352** | **0.389** | 0.358 | 0.392 | 0.361 | 0.391 |
| | 336 | 0.385 | 0.417 | **0.372** | **0.403** | 0.376 | 0.407 | 0.378 | 0.411 |
| | 720 | 0.421 | 0.446 | **0.409** | **0.434** | 0.413 | 0.442 | 0.416 | 0.442 |
| | Avg. | 0.368 | 0.401 | **0.354** | **0.391** | 0.357 | 0.395 | 0.361 | 0.397 |
| PEMS04 | 12 | 0.112 | 0.228 | **0.096** | **0.211** | 0.102 | 0.213 | 0.108 | 0.219 |
| | 24 | 0.141 | 0.258 | **0.130** | **0.242** | 0.134 | 0.245 | 0.135 | 0.247 |
| | 36 | 0.187 | 0.286 | **0.165** | **0.279** | **0.165** | **0.279** | 0.172 | 0.283 |
| | 48 | 0.204 | 0.313 | **0.187** | **0.301** | 0.189 | 0.304 | 0.193 | 0.307 |
| | Avg. | 0.161 | 0.271 | **0.145** | **0.258** | 0.148 | 0.260 | 0.152 | 0.264 |
| Linear Probing | | Random Init. | | TimeDART | | SimMTM | | TimeMAE | |
| Metrics | | Acc | F1 | Acc | F1 | Acc | F1 | Acc | F1 |
| HAR | | 0.8542 | 0.8578 | **0.8976** | **0.9005** | 0.8732 | 0.8756 | 0.8858 | 0.8862 |

Table 22: Comparison of parameters and training times on Traffic dataset.

| | Methods | Params | Training Time/per epoch |
|---|---|---|---|
| SSL. | TimeDART | 2.36M (pre-training) / 2.16M (fine-tuning) | 510s (pre-training) / 349s (fine-tuning) |
| | SimMTM | 14.5M (pre-training) / 2.16M (fine-tuning) | 85min (pre-training) / 349s (fine-tuning) |
| | TimeMAE | 1.13M (pre-training) / 2.16M (fine-tuning) | 91min (pre-training) / 349s (fine-tuning) |
| | CoST | 2.66M(pre-training) / 2.16M(fine-tuning) | 24mins(pre-training) / 349s(fine-tuning) |
| SL. | PatchTST | 4.27M | 158s |

