# OpenReview forum: "TimeDART: A Diffusion Autoregressive Transformer for Self-Supervised Time Series Representation"
_ICML.cc/2025/Conference — ICML 2025 poster_

### Official Review · Reviewer_S1wT · 2025-03-13

**Overall Recommendation:** 4

**Summary:**

This paper introduces TimeDART, a self-supervised time series representation learning framework that integrates autoregressive modeling with a denoising diffusion process. The method consists of a causal Transformer encoder with a patch-based embedding strategy to capture global trends, while a denoising diffusion process refines fine-grained local patterns. The authors claim that this combination improves transferability and representation quality for downstream tasks. Empirical evaluations on nine publicly available datasets demonstrate the effectiveness of TimeDART, outperforming various state-of-the-art baselines in time series forecasting and classification.

**Claims And Evidence:**

In this work, the authors hold that combination of auto-regressive and diffusion optimization schemes can be used to obtain transferrable time series representation so as to benefit the target downstream tasks. The specific experimental results are provided in the experimental part.

**Essential References Not Discussed:**

No.

**Experimental Designs Or Analyses:**

The experimental design is mostly sound. The main results of the proposed methods and compared baselines, the ablation of this model et al are discussed in the experimental parts.

**Methods And Evaluation Criteria:**

Yes, the chosen benchmarks (ETT, Electricity, Traffic, Weather, PEMS, EEG, Epilepsy, HAR) are widely used in time series forecasting and classification. The evaluation settings are appropriate:

**Other Comments Or Suggestions:**

No.

**Other Strengths And Weaknesses:**

Strengths:
1. Novel integration of diffusion models and autoregressive modeling for time series representation learning.
2. Extensive empirical validation across nine datasets for forecasting and classification.
3. Strong performance against self-supervised and supervised baselines, demonstrating robustness.
4. Cross-domain evaluation, showing adaptability across different time series applications.
5. Comprehensive ablation studies, highlighting the contributions of different components.

Weakness:
1. Baseline selection could be improved—how does TimeDART compare to LLM-based time series models or non-diffusion self-supervised methods?
2. I Suggest the author further improve the writing details to improve the impact of this paper.

**Questions For Authors:**

Q1: Whether the pre-trained time series model can be fine-tuned in a new manner? For example, it obeys the same training procedure as the pre-training stage.

Q2: The authors combine two types of prevalent generative optimization paradigm together for self-supervised time series representation. Can you describe the great difference between this two training schemes in terms of time series representation?

**Relation To Broader Scientific Literature:**

The paper advances self-supervised time series representation learning by integrating autoregressive modeling with denoising diffusion, addressing limitations in masked modeling (e.g., TimeMAE) and contrastive learning (e.g., TS2Vec). It extends diffusion models beyond probabilistic forecasting (e.g., TimeGrad, CSDI) to representation learning, improving both global trend and local pattern capture. A comparison with large-scale time series foundation models (e.g., Chronos, TimesFM et al) would further clarify its positioning in the field.

**Theoretical Claims:**

The paper primarily relies on empirical validation rather than formal theoretical proofs. The diffusion loss formulation aligns with standard ELBO principles and appears correct. While the integration of autoregressive modeling and diffusion is well-motivated, a more detailed theoretical justification could further strengthen the claims.

---

> ### Author Rebuttal · Authors · 2025-03-29
>
> > All full results are at https://anonymous.4open.science/r/TimeDART-results-ECBD/
> >
> > [A1 for Q1]--->para4,para5, [A3 for W1] —> para3
>
> **A1 for Q1**
>
> Please refer to `A1 for Q1` in our rebuttal to the second Reviewer TNxq.
>
> **A2 for Q2**
>
> Of course, we are willing to elaborate on our understanding of these two generative paradigms in the context of time series representation learning, and we welcome your corrections if there are any inaccuracies:
>
> From the perspective of the characteristics of the data itself, we believe that time series data possesses traits of both language data and image data. First, language data generally has high information density and exhibits strong contextual dependencies. Autoregressive modeling methods share a natural alignment with human language [1], and similarly, time series data emphasizes dependencies from past to current states. Secondly, image data typically has lower information density [2] and places greater emphasis on jointly modeling global spatial relationships and locality. Time series data also shares similar characteristics.
>
> Recently, there have been excellent works applying diffusion to autoregressive language models, such as LlaDA [3], as well as works applying autoregressive approaches to image generation, such as VAR [4]. These are inspiring contributions. As contemporaneous work, we carefully examined the distinct roles that diffusion modeling and autoregressive modeling play in time series representation learning.
>
> On one hand, autoregressive modeling can capture relationships from left to right. However, we recognize that time series data also shares characteristics with image data, including locality features. Due to its relatively low information density, it is challenging to effectively fit time series data using autoregressive methods without employing discretization techniques like VQVAE.
>
> On the other hand, diffusion models can effectively model locality features in time series, such as abrupt weather changes within a single day or week. However, modeling time series requires a significant focus on trend modeling [5],[6]. otherwise, the model risks overfitting to drift.
>
> This is the source of our motivation: by combining these two approaches, we can capture both long-term dynamic evolution and subtle local patterns in a unified manner .
>
> [1] GPT-1: Improving Language Understanding by Generative Pre-Training
>
> [2] MAE: Masked Autoencoders Are Scalable Vision Learners
>
> [3] LlaDA: Large Language Diffusion Models
>
> [4] VAR: Visual Autoregressive Modeling: Scalable Image Generation via Next-Scale Prediction
>
> [5] Autoformer: Decomposition Transformers with Auto-Correlation for Long-Term Series Forecasting
>
> [6] FDF: Flexible Decoupled Framework for Time Series Forecasting with Conditional Denoising and Polynomial Modeling
>
> **A3 for W1**
>
> For non-diffusion self-supervised methods, comparisons are provided in Tables 2, 3, and 4 of the paper, including methods based on mask modeling and contrastive learning. Specific baseline details can be reviewed in Section 4.1 (Baselines) of the article. The experiments demonstrate that, compared to the baseline methods we selected, TimeDART achieves superior downstream fine-tuning performance after pre-training.
>
> For LLM-based methods, please refer to `A2 for Q2` in our rebuttal to the first Reviewer baxN where we compare TimeDART with LLM-based methods like UniTime.
>
> **A4 for W2**
>
> Thank you for your sincere suggestions. We will continue to refine our paper based on the feedback from all reviewers, especially the analysis regarding the differences between autoregressive and diffusion modeling that you mentioned in question 2.
>
> Thank you for your encouraging score. We sincerely look forward to your reply.

---

### Official Review · Reviewer_6U5m · 2025-03-13

**Overall Recommendation:** 4

**Summary:**

This paper presents a self-supervised time series representation learning method. It combines autoregressive modeling with the denoising diffusion process. Key ideas involve normalizing and patch-embedding data, using a causal Transformer encoder for long-term evolution and a patch-level diffusion/denoising mechanism for local patterns. Results show TimeDART outperforms baselines in forecasting and classification tasks, both in-domain and cross-domain.

**Claims And Evidence:**

most of the claims are supported by clear and convincing evidence.

**Essential References Not Discussed:**

No

**Experimental Designs Or Analyses:**

the forecasting experiments are well-configured and analysed

**Methods And Evaluation Criteria:**

The methods and evaluation criteria are sensible.

**Other Comments Or Suggestions:**

n/a

**Other Strengths And Weaknesses:**

strengths:

(1) Demonstrating that autoregressive methods can be applied to self-supervised tasks and that diffusion models can explicitly introduce noise and model local patterns, which offers new insights to the forecasting

(2) The paper is well-designed and the presentation is clear.

(3) The technics of combination of the autoregressive  and the denoising diffusion process seems sound.

Weaknesses:

(1) For self-supervised learning, one of the significant advantages lies in its performance after few-shot fine-tuning in downstream tasks. I'm curious about how TimeDART would perform when only 5% or 10% of the samples are used for downstream fine-tuning

(2) The transition from the description of the reverse process in Equation (7) to the elaboration of the optimization objective in Equation (8) seems rather large. The authors should provide a more detailed derivation of the optimization objective

(3) add a complete table to illustrate the specific model and training parameters for each dataset.

**Questions For Authors:**

see weaknesses

**Relation To Broader Scientific Literature:**

The model addresses the challenges of traditional self-supervised methods, offering an approach with improved performance in time series forecasting

**Theoretical Claims:**

yes, equations follow basic mathematical operations and seem logically sound.

---

> ### Author Rebuttal · Authors · 2025-03-29
>
> > All full results are at https://anonymous.4open.science/r/TimeDART-results-ECBD/
> >
> > [Table1]--->para7, [Table2]--->para8
>
> **A1 for Q1**
>
> We conduct detailed few-shot experiments on the performance of the model with 5% or 10% fine-tuning data, including forecasting and classification tasks. The results can be found in the [Table1] below.
>
> [Table1]
>
> |Portion|5%|10%|100%|Random Init|
> |-|-|-|-|-|
> |Metrics|MSE/MAE|MSE/MAE|MSE/MAE|MSE/MAE|
> |ETTh2|0.356/0.392|0.354/0.391|0.346/0.387|0.358/0.396|
> |PEMS04|0.142/0.254|0.140/0.250|0.133/0.245|0.145/0.255|
> ||Acc/F1|Acc/F1|Acc/F1|Acc/F1|
> |HAR|0.8843/0.8901|0.9014/0.9053|0.9247/0.9286|0.8738/0.8723|
>
> **A2 for Q2**
>
> The following is our detailed derivation from formula [7] to formula [8], which we will integrate into the paper in the future to highlight the comprehensiveness and rigor of the paper.
>
> The original theoretical loss should be in the form of cross entropy:
>
> $$
> L_{ideal}=H(p_\\theta(x^0\_{1:j}),q(x^0_{1:j}))=\\mathbb{E}_{q(x^0\_{1:j})}(-\\log p\_\\theta(x^0\_{1:j}))
> $$
>
> Use the Evidence Lower Bound (ELBO) method and ignore the patch index subscript:
>
> $$
> \begin{align}
> L\_{ideal} &\\leq \\mathbb{E}\\_{q(x^0)} \\left(\mathbb{E}\_{q(x^{1:T}|x^0)}\\left[\\log \\frac{q(x^{1:T}|x^0)}{p\_\\theta(x^{0:T})}\\right]\\right) \\\\
> &=\\mathbb{E}\_{q(x^{0:T})} \\left[\\log \\frac{q(x^{1:T}|x^0)}{p\_\\theta(x^{0:T})}\\right] \\\\
> &:= L\_{diff}
> \end{align}
> $$
>
> Expand and use the Bayes formula:
>
> $$
> \\begin{align}
> L &= \\mathbb{E}\_{q(x^{0:T})} \\left[ \\log \\frac{q(x^{1:T}|x^0)}{p\_\\theta(x^{0:T})} \\right] \\\\
> &= \\mathbb{E}\_{q(x^{0:T})} \\left[ \\log \\frac{\\prod\_{s=1}^T q(x^s|x^{s-1})}{p(x^T) \\prod\_{s=1}^T p\_\\theta(x^{s-1}|x^s)} \\right] \\\\
> &= \\mathbb{E}\_{q(x^{0:T})} \\left[ \\log \\frac{ q(x^T|x^0)}{p(x^T)} + \\sum\_{s=2}^T \\log \\frac{q(x^{s-1}|x^s, x^0)}{p\_\\theta(x^{s-1}|x^s)} - \\log p\_\\theta(x^0|x^1) \\right] \\\\
> \\end{align}
> $$
>
> The first term is a constant, the third term is the reconstruction loss, and only the second term is discussed:
>
> $$
> \\begin{align}
>     \\mathbb{E}\_{q(x^{0:T})} \\sum\_{s=2}^T \\log \\frac{q(x^{s-1}|x^s, x^0)}{p\_\\theta(x^{s-1}|x^s)} &= \\int\\text{d}x^{0:T}~ q(x^{0:T})\\cdot\\sum\_{s=2}^T \\log \\frac{q(x^{s-1}|x^s, x^0)}{p\_\\theta(x^{s-1}|x^s)} \\\\
>     &=\\sum\_{s=2}^T\\mathbb{E}\_{q(x^0,x^s)}\\left[D_{KL}(q(x^{s-1}|x^s, x^0)\\|p\_\\theta(x^{s-1}|x^s))\\right]
> \\end{align}
> $$
>
> Applying Bayes' theorem again:
>
> $$
> q(x^{s-1} | x^s, x^0) = \\mathcal{N}(x^{s-1},\\tilde{\\mu}_t(x^s,x^0),\\tilde{\\beta}_s I)
> $$
>
> It can be assumed that the predicted distribution can also be expressed as:
>
> $$
> p_\\theta(x^{s-1}|x^s) = \\mathcal{N}(x^{s-1};\\mu_{\\theta}(x^s,s),\\sigma_s^2 I)
> $$
>
> According to the KL divergence of two Gaussian distributions with the same variance, we can get:
>
> $$
> L\_{diff}=\\mathbb{E}\_{q(x^0,x^s)}\\left[\\frac{1}{2\\sigma\_s^2} ||\\tilde{\\mu\_s} (x^s, x^0)-\\mu\_\\theta(x^s,s) ||^2\\right]
> $$
>
> Since $\\tilde{\\mu}_s$ and $x^0$ are linearly related, we turn to predict the original space, so:
>
> $$
> L_{diff}\\sim \\mathbb{E}_{\\epsilon, q(x^0)} \\left[ || x^0 - x^{out}  ||^2 \\right].
> $$
>
> Adding the cumulative sum of subscripts and the notation from the original paper, we get:
>
> $$
> L_{ours} =  \sum_{j=1}^{N} \\mathbb{E}_{\\epsilon, q(x_j^0)} \\left[ || x_j^0  - g(\hat{z_j}^{in} - f(z\_{1:j-1}^{in}) ||^2 \\right]
> $$
>
> **A3 for Q3**
>
> Below is a table of the key parameters for the complete model, pre-training, and fine-tuning process, where `p1/p2/…` means we searched for these parameters (such as d_model, learning_rate), or we dynamically modified these parameters based on the size of the dataset (such as batch_size):
>
>
> [Table2]
>
> |Tasks|Encoder||Decoder||Pre-training|||Fine-tuning||||
> |-|-|-|-|-|-|-|-|-|-|-|-|
> ||e_layers|d_model|d_layers|d_model|learning_rate|batch_size|epoches|learning_rate|lr_scheduler|batch_size|epoches|
> |Forecasting|2|8/32/128/512|1|encoder_d_model|0.001,0.0005,0.0001|8,16|30,50|0.001,0.0005,0.0001|cosine/exponential decay|8,16,32,64|10|
> |Classification|2|64/128/256|1|encoder_d_model|0.001,0.0005,0.0001|16,64,128|30,50|0.001,0.0005,0.0001|cosine/exponential decay|16,64,128|10|
>
> Thank you for your encouraging score. We sincerely look forward to your reply.

---

### Official Review · Reviewer_TNxq · 2025-03-13

**Overall Recommendation:** 3

**Summary:**

The paper introduces TimeDART, a novel self-supervised learning framework for time series analysis that integrates autoregressive modeling with diffusion-based denoising. The framework aims to address the limitations of existing methods, such as masked autoencoders, contrastive learning, and autoregressive approaches, particularly their susceptibility to noise. TimeDART employs a causal Transformer encoder and a cross-attention-based denoising mechanism to capture both global dynamics and local patterns in time series data. The paper demonstrates the effectiveness of TimeDART through extensive experiments, showing improvements in both time series prediction and classification tasks.

## update after rebuttal
The authors have actively responded to reviewers' comments. Though some of my concern about the utilization of diffusion model still remains, to some extent, they've empirically illustrated that it worked from the view of self-supervised learning.

**Claims And Evidence:**

The claims in the paper are clear and well-supported by evidence. The authors effectively argue that combining autoregressive modeling with diffusion denoising helps capture both global and local patterns in time series data. The use of diffusion denoising is justified to mitigate the overfitting problem of autoregressive models to noise and anomalies, and the cross-attention decoder is introduced to address the local dependence issue of diffusion models.

**Essential References Not Discussed:**

There are no essential references missing from the discussion.

**Experimental Designs Or Analyses:**

The experimental designs and analyses are sound. However, there is a concern about the consistency of hyperparameters across different methods, as the results of some baseline methods (e.g., PatchTST) appear weaker than those reported in their original papers.

**Methods And Evaluation Criteria:**

The methods and evaluation criteria are appropriate for the problem.

**Other Comments Or Suggestions:**

None.

**Other Strengths And Weaknesses:**

Strengths:
* The writing is clear, and the code is provided, which enhances the reproducibility of the results.
* The experiments are well-designed and demonstrate the superiority of the proposed method.

Weaknesses:
* The combination of autoregressive and diffusion models is straightforward. Both techniques are commonly used in many other fields.
* The combination of autoregressive modeling and diffusion denoising is only used during the pretraining stage, and a different paradigm is adopted for downstream tasks, which weakens the innovation of the approach.
* The runtime efficiency of the algorithm is not reported, raising concerns about the computational cost of combining autoregressive modeling with multiple denoising steps.

**Questions For Authors:**

1. Why do the authors abandon autoregressive modeling and diffusion in the downstream task, given that these methods are central to their claims about capturing global and local patterns? What’s the performance of keeping autoregressive modeling and diffusion model in forecasting task?
2. If the denoising decoder is removed during fine-tuning, is the diffusion model still necessary? Could other simpler backbone networks achieve similar results?
3. What is the runtime efficiency of the algorithm, especially given the potential computational cost of combining autoregressive modeling with multiple denoising steps?
4. I noticed that in the data factory script, the setting of drop_last is True (/data_provider/data_factory.py#L41). This will lead to wrong results of the whole experiment [1].
5. What are the effects of different lookback windows on the results?

[1] TFB: Towards Comprehensive and Fair Benchmarking of Time Series Forecasting Methods.

**Relation To Broader Scientific Literature:**

The main contribution of the paper lies in the combination of autoregressive and diffusion models in self-supervised learning for time series analysis. This approach addresses the overfitting problem in autoregressive methods and enhances the ability to capture local information through diffusion models.

**Theoretical Claims:**

The theoretical claims are supported by a statistical invariant property analysis and a rough derivation of the optimization objective provided in the appendix. There are no significant issues with the theoretical foundations of the paper.

---

> ### Author Rebuttal · Authors · 2025-03-29
>
> > All full results are at https://anonymous.4open.science/r/TimeDART-results-ECBD/
> >
> > [Table1]--->para4,para5, [Table2]--->para6, [A3 for Q3] --->para1, [A5 for Q5]--->para2
>
>
> **A1 for Q1**
>
> We abandon autoregressive modeling and diffusion in the downstream task based on three considerations: First, we not only performed forecasting tasks but also discriminative tasks such as classification, which are difficult to directly adapt to this paradigm. Second, almost all evaluations of self-supervised methods for time series follow the approach of transferring only the encoder and attaching different heads for various downstream adaptations (e.g., TimeSiam, SimMTM). Therefore, to enable comparisons and ensure fairness, we adhered to the same transfer method. Lastly, autoregressive inference and diffusion sampling would be extremely slow.
>
> Early of this work, we have designed a downstream denoising autoregressive paradigm for forecasting tasks and conducted experiments. However, based on the considerations above, we decided not to use this approach further. We appreciate the reviewer's question, as it allows us to revisit our initial design. The results can be found in the [Table1] below.
>
> [Table1]
> The following table shows the average of predicted window on [96,192,336,720]. Detailed designs can be found at para5 in link above. The results show that retaining the decoder for prediction can achieve similar results.
>
> |Methods|Diff+non AR|Diff+non AR Random Init|Diff+AR|Diff+AR Random Init|TimeDART|Random Init|
> |-|-|-|-|-|-|-|
> |Metrics|MSE/MAE|MSE/MAE|MSE/MAE|MSE/MAE|MSE/MAE|MSE/MAE|
> |ETTh2|0.352/0.389|0.363/0.397|0.359/0.394|0.372/0.400|**0.346/0.387**|0.358/0.396|
> |Exchange|0.346/0.406|0.383/0.442|**0.339/0.385**|0.366/0.422|0.359/0.405|0.440/0.450|
>
> **A2 for Q2**
>
> Even though the denoising decoder is removed during fine-tuning, it remains **necessary**, as the ablation experiments show: removing the diffusion model during pre-training leads to a decline in performance for both downstream forecasting and classification tasks.
>
> We can use a simpler MLP as a decoder. For simplicity, we directly use the concat strategy to concatenate the output of the causal encoder and the embedding of the noise added patches in the dim axis. In order to keep the parameters consistent with the original 1-layer Transformer (about $12d_{model}^2$), we use a Squencial(Linear($2d_{model}$, $4d_{model}$), ReLU, Linear($4d_{model}$, $d_{model}$) ) MLP as the denoising decoder. The results can be found in the [Table2] below.
>
> [Table2]
> The following table shows the average of predicted window on [96,192,336,720]. The results show that simpler's MLP is slightly weaker than TRM, but almost the same.
>
> |Decoder|MLP|TRM|Random Init|
> |-|-|-|-|
> |Metrics|MSE/MAE|MSE/MAE|MSE/MAE|
> |ETTh2|0.347/0.387|0.346/0.387|0.358/0.396|
> |PEMS04|0.134/0.245|0.133/0.245|0.145/0.255|
> ||Acc/F1|Acc/F1|Acc/F1|
> |HAR|0.9197/0.9186|0.9247/0.9286|0.8738/0.8723|
>
> **A3 for Q3**
>
> Please refer to `A1 for Q1, paragraph 1` in our rebuttal to the first Reviewer baxN, where the table of gpu memory and calculation is given.
>
> **A4 for Q4**
>
> You have raised an important point, however, we must clarify the entire data processing and experimental evaluation section:
>
> You mentioned that in `/data_provider/data_factory.py#L41`, setting `drop_last` to `True` might lead to discarding some samples, potentially biasing the results. However, at #L44 and #L45, when the downstream task is forecasting, the `batch_size` is set to 1. With this, no forecasting test samples will be dropped even `drop_last` is set to `True`. Furthermore, at #L42 and #L43, when the downstream task is classification, although `batch_size` is set to `args.batch_size`, at #L69 and #L70 `drop_last` is explicitly reset to `False`. Thus, the classification task is also unaffected.  For the sake of rigor: we double-checked the code and printed the key parameters. The results confirm that the critical parameters are as described above.
>
> **A5 for Q5**
>
> Please refer to `A1 for Q1, paragraph 3` in our rebuttal to the first Reviewer baxN, where the table of different look-back windows is given.
>
> **A6 for W1**
>
> As mentioned in `a3 for w1` in our rebuttal to first Reviewer baxN, we did not mechanically combine these two techniques in pre-training. Instead, we approached the uniqueness of time series data with careful consideration. Simply using autoregressive optimization during pre-training does not yield satisfactory results when transferred to downstream tasks. However, through our novel approach of explicitly introducing noise, we found that the pre-training performance indeed improved, which inspired the development of our current complete work.
>
> **A6 for W2**
>
> Please refer to `A1 for Q1`.
>
> **A7 for W3**
>
> Please refer to `A3 for Q3` above or `A1 for Q1, paragraph 1` in our rebuttal to the first Reviewer baxN.
>
> Your score is very important to us. We hope we can solve your questions. We sincerely look forward to your reply.

---

> > ### Comment · Reviewer_TNxq · 2025-04-06
> >
> > Thank you for your response. Some of the concerns still remain.
> >
> > Q1 for A1:
> > If it is merely a process of adding and removing noise during training, it can hardly be called "diffusion". Or rather, this is just a simple denoising autoencoder. Literally speaking, **this may conflict with the core contribution of your method**.
> >
> > Moreover, it further raises the question about the effect of the self-supervised pretraining. I’ve noticed that the encoder has been adapted to the downstream tasks via fine-tuning, as stated in Line 224-235 of Sec. 3.3 in your paper. However, the effect of pretraining cannot be well understood via the experiments and ablations study. Could you please give any explanation or analysis on it?
> >
> > Q2:
> > Regarding the diffusion part of your model, you believe it captures local information. Do you have any insights or visual analysis to prove the effectiveness of the noise adding process in capturing local temporal features? What are the advantages of this noise adding method compared to other methods for capturing local features, such as the duel attention in Pathformer [1]?
> >
> > [1] Pathformer: Multi-scale Transformers with Adaptive Pathways for Time Series Forecasting

---

> > > ### Author Response · Authors · 2025-04-08
> > >
> > > > Full results are at https://anonymous.4open.science/r/TimeDART-results-ECBD/
> > > >
> > > > [Table1] —>para9
> > >
> > > **A1 for Q1**
> > >
> > > We understand the reviewer's concerns and would like to clarify: our method is not a traditional denoising autoencoder. Our core contribution lies in proposing a unified pre-training framework that integrates autoregressive modeling and denoising diffusion models. Below we elaborate on the differences between TimeDART and traditional DAE:
> > >
> > > 1. We use cosine scheduled multi-step noising and single-step denoising to train the representation network and denoising decoder, and use the autoregressive diffusion loss as the optimization target, which is consistent with the classic diffusion architecture.
> > >
> > > 2. The classic denoising autoencoder adds noise to the input data so that the network learns the corrupted data. It does not involve the autoregressive information as a condition, the complex noise addition mechanisms and diffusion loss optimization theory.
> > >
> > > 3. Although we removed the denoising decoder in the downstream task to maintain consistency and fairness with the baseline, this does not mean that our denoising decoder has no denoising capability in downstream tasks. As we mentioned in `A1 for Q1` of the first rebuttal, we can also use the denoising decoder to fine-tune, and the performance is basically the same as the method in the paper.
> > >
> > >
> > > As noted in [1], only a few components of diffusion models are essential for learning good representations, while many others are not essential. TimeDART, tailored for time-series data, retains these key elements while adapting to the unique characteristics of such data, consistent with our core contributions.
> > >
> > > Regarding the effect of self-supervised pre-training, we evaluate it along two dimensions, each supported by corresponding experiments:
> > >
> > > 1. **End-to-end pre-train fine-tune vs. random initialization**
> > >
> > >    This is a widely-used evaluation approach comparing models with pre-trained encoders against those randomly initialized (SimMTM, TimeSiam).  In Sec. 4.1, the "TimeDART" and "Random Init" columns in our experimental tables consistently show that TimeDART outperforms random initialization, highlighting the value of our pre-training framework.
> > >
> > >    To further address your concerns, we conducted experiments under another evalution approach: **linear probing**, where the pre-trained encoder is fixed, and only the newly added task-specific projector is fine-tuned. Similarly, we compared this with a randomly initialized encoder, also fixed during fine-tuning. The results is in [Table1].
> > >
> > > 2. **Ablation studies on key components** :
> > >
> > >    We have performed detailed ablation experiments on TimeDART's two main components in Sec4.3: the autoregressive mechanism and the denoising diffusion process. For the autoregressive component, we removed causal masks and related elements during pre-training. For the diffusion process, we eliminated the diffusion module and denoising decoder. Removing either or both components degraded downstream performance, often falling below random initialization, validating the importance of our design choices.
> > >
> > > [Table1] Linear probing fine-tuning for TimeDART. MSE/MAE for ETTh2 and PEMS04, Acc/F1 for HAR.
> > >
> > > |Linear Probing|Random Init|TimeDART|SimMTM|TimeMAE|
> > > |-|-|-|-|-|
> > > |ETTh2|0.368/0.401|**0.354/0.391**|0.357/0.395|0.361/0.397|
> > > |PEMS04|0.161/0.271|**0.145/0.258**|0.148/0.260|0.152/0.264|
> > > |HAR|0.8542/0.8578|**0.8976/0.9005**|0.8732/0.8756|0.8858/0.8862|
> > >
> > > As shown in the table, under linear probing, TimeDART maintains the same trend as the main experiment, outperforming random initialization and the article's baselines.
> > >
> > > [1] Chen, X., Liu, Z., Xie, S., & He, K. (2024). Deconstructing denoising diffusion models for self-supervised learning. ICLR 2025
> > >
> > > **A2 for Q2**
> > >
> > > While patch-based methods(PatchTST, Pathformer) excel at local feature extraction, they often struggle with inherent noise. Building on this, we use patch-level embeddings and introduce an explicit noising-denoising process during pre-training, optimized with diffusion loss in the original space. This enhances the encoder's robustness to noise, with ablation studies confirming its effectiveness for downstream tasks.
> > >
> > > We have done preliminary T-SNE visualization analysis after pre-training with and without noise, and observed that different datasets exhibit more clear clustering-like patterns after adding noise. If lucky enough to be accepted, we would add the visualization to the final submitted camera ready version.
> > > Compared to other patch-based methods, our approach offers key advantages for capturing local features:
> > > 1. Unified modeling : Unlike methods focused solely on local features, our framework integrates both long-term dependencies and local patterns, paving the way for a foundation model for time series.
> > > 2. Patch-based compatibility : Our noising-denoising process can be seamlessly applied to other patch-based models (e.g., Pathformer) without extensive modifications.

---

### Official Review · Reviewer_baxN · 2025-03-13

**Overall Recommendation:** 4

**Summary:**

Authors propose a novel self-supervised time series representation pre-training framework that integrates two popular generative paradigms to enhance representation transferability. Specifically, they employ a causal Transformer encoder for autoregressive prediction while incorporating a denoising diffusion process to recover fine-grained local patterns. Extensive experiments on time series classification and forecasting tasks validate the effectiveness of the proposed approach.

**Claims And Evidence:**

The claims in the paper are well-supported by empirical evidence. TimeDART’s effectiveness in capturing both global and local sequence features is validated through strong performance across multiple benchmark datasets. Extensive experiments in forecasting and classification, along with ablation studies, demonstrate its advantages over state-of-the-art methods. The integration of autoregressive modeling and denoising diffusion is shown to enhance representation learning, confirming the validity of the proposed approach.

**Essential References Not Discussed:**

None

**Experimental Designs Or Analyses:**

The experimental design is well-structured and comprehensive. The authors evaluate TimeDART across diverse benchmark datasets using appropriate metrics (MSE, MAE, accuracy, F1-score). Ablation studies effectively validate the contributions of autoregressive modeling and the denoising diffusion process. The inclusion of both in-domain and cross-domain evaluations further strengthens the robustness of the findings.

**Methods And Evaluation Criteria:**

The proposed methods and evaluation criteria are well aligned with the problem. TimeDART effectively integrates autoregressive modeling and denoising diffusion for self-supervised time series representation learning. The use of diverse benchmark datasets across forecasting and classification tasks ensures a comprehensive evaluation. Metrics such as MSE, MAE, accuracy, and F1-score are appropriate, and ablation studies further validate the contributions of key components.

**Other Comments Or Suggestions:**

See above

**Other Strengths And Weaknesses:**

Strengths
1. The paper effectively combines autoregressive modeling with a denoising diffusion process, offering a novel perspective to self-supervised time series representation learning.
2. The method is extensively validated across multiple benchmark datasets for both forecasting and classification tasks, demonstrating consistent performance gains over state-of-the-art baselines.
3. The paper includes detailed ablation studies and hyperparameter sensitivity analysis, reinforcing the contributions of key components and ensuring the method's robustness.

Weakness
1. While the empirical results are strong, it would be better if the paper could provide a formal theoretical analysis explaining why the combination of autoregressive modeling and diffusion improves representation learning.
2. It seems that the paper does not provide insights into the computational cost of TimeDART compared to other self-supervised methods, especially regarding training efficiency and scalability.

**Questions For Authors:**

Question 1: Diffusion-based models often introduce additional computational overhead due to iterative denoising steps. How does TimeDART compare in terms of training and inference time relative to standard self-supervised methods like contrastive learning or masked autoencoders? Have you evaluated the scalability of TimeDART on longer sequences or larger datasets? A comparison of computational trade-offs would clarify its practical applicability.

Question 2: Recent advancements in time series modeling include large-scale pre-trained models (e.g., TimeGPT) that generalize across domains. How does TimeDART compare in terms of representation quality and transferability to such foundation models? Have you considered evaluating it on cross-domain tasks beyond the current benchmark datasets? Addressing this would help position TimeDART within the broader landscape of time series pre-training.

**Relation To Broader Scientific Literature:**

The paper builds on prior work in self-supervised time series representation learning, integrating autoregressive modeling and diffusion-based denoising. It extends masked modeling and contrastive learning approaches by introducing a hybrid generative framework. Compared to traditional autoregressive methods, TimeDART mitigates error accumulation, while diffusion models typically used for probabilistic forecasting are repurposed for representation learning, bridging gaps in existing time series pre-training strategies.

**Theoretical Claims:**

The paper's theoretical foundations are consistent with established principles, particularly in diffusion-based modeling and autoregressive learning. The diffusion loss formulation aligns with ELBO principles, and its application appears correct. While a deeper theoretical justification could further enhance clarity, the empirical results strongly support the proposed approach.

---

> ### Author Rebuttal · Authors · 2025-03-29
>
> > All full results are at https://anonymous.4open.science/r/TimeDART-results-ECBD/
> >
> > [Table1] —>para1,[Table2] —>para2,[Table3] —> para3
>
> **A1 for Q1**
> 1. For efficiency, we used a lightweight denoising decoder during pre-training.  After training, similar to masked autoencoders, only the embedding layer and encoder network are transferred for downstream tasks. As a result, TimeDART introduces small additional computational overhead in terms of GPU memory usage and computation speed. Specific comparative results can be found in the [Table1] below.
> 2. In forecasting tasks, we have evaluated performance on large datasets especially traffic and pems. Dataset details, including variables and lengths, are in Section 4.1. We also pre-trained on all six datasets combined, reaching approximately 43M training time points (channel independence). This data volume is significantly larger than the baseline in the article.
> 3. To evaluate the scalability of TimeDART for longer sequences, we conducted additional experiments with different look-back window sizes, especially extending it to 576 and 720 in ETTh2. We observed a further reduction in prediction MSE, indicating that TimeDART scales well for longer sequences. The detailed results can be found in the [Table2] below.
>
> [Table1]
> The data in the following table is recorded in the Traffic dataset with a look-back window=336 and a predicted window=336.
> |Methods|Params|Training Time/per epoch|
> |-|-|-|
> |TimeDART|2.36M(pt)/2.16M(ft)|510s(pt)/349s(ft)|
> |SimMTM|14.5M(pt)/2.16M(ft)|85mins(pt)/349s(ft)|
> |TimeMAE|1.13M(pt)/2.16M(ft)|91mins(pt)/349s(ft)|
> |Cost|2.66M(pt)/2.16M(ft)|24mins(pt)/349s(ft)|
> |PatchTST(supervised)|64.32M|158mins|
>
> [Table2]
> The following table shows the average of predicted window on [96,192,336,720].
> |look-back|TimeDART|Random Init|
> |-|-|-|
> |ETTh2|MSE/MAE|MSE/MAE|
> |96|0.373/0.398|0.384/0.407|
> |192|0.364/0.391|0.374/0.401|
> |336|0.346/0.387|0.358/0.396|
> |576|0.343/0.384|0.355/0.392|
> |720|0.339/0.379|0.352/0.388|
> |PEMS04|MSE/MAE|MSE/MAE|
> |96|0.133/0.245|0.145/0.255|
> |192|0.130/0.240|0.140/0.252|
> |336|0.125/0.235|0.134/0.247|
>
> **A2 for Q2**
> 1. Due to limitations in GPUs and the time constraints of the rebuttal process, and given the significant disparity in the scale of training data, it is challenging for us to directly compare with large foundation models in a short period. To address your concerns regarding representation quality and transferability, we have instead conducted some comparisons with SOTA methods based on LLMs (e.g. UniTime, GPT4TS), which we hope will help alleviate your doubts and concerns.  Specific comparative results can be found in the [table3] below.
> 2. Regarding the cross-domain issue, we mixed all in-domain datasets for general pre-training in forecasting tasks and fine-tuned across domains. As noted in `a1 for q1`, this yielded approximately 43M training time points. Experiments (Section 4.2) demonstrate that TimeDART’s cross-domain representation learning, under large-scale pre-training, shows improvement compared to no pre-training.  As for going beyond the current benchmark datasets, we are considering conducting general pre-training on larger public datasets (e.g., UTSD) to further validate TimeDART’s representation ability.
>
> [Table3]
> The look-back window is reset to 96 same to UniTime to ensure fair comparison. Model is pretrained on ETT(4subsets), Exchange, Electricity, Traffic datasets. The following table shows the average of predicted window on [96,192,336,720]. Experiments show that TimeDART can still learn better representations compared to LLM-based methods.
>
> |Models|TimeDART|Random Init|Unitime|GPT4TS|PatchTST|
> |-|-|-|-|-|-|
> |Metrics|MSE/MAE|MSE/MAE|MSE/MAE|MSE/MAE|MSE/MAE|
> |ETTh2|**0.376/0.398**|0.384/0.404|0.378/0.403|0.386/0.406|0.398/0.416|
> |ETTm2|**0.287/0.333**|0.301/0.350|0.293/0.334|0.321/0.356|0.340/0.373|
> |Exchange|**0.361**/0.406|0.389/0.424|0.364/**0.404**|0.421/0.446|0.411/0.444|
> |Electricity|**0.200/0.293**|0.212/0.303|0.216/0.305|0.251/0.338|0.221/0.311|
>
> **A3 for W1**
> Due to time constraints, we just provide a brief explanation. Our assumption is that traditional autoregressive pre-training excels in capturing long-term dependencies but struggles with inherent time series noise, which is hard to avoid. By introducing noise into the pre-training process, we encourage the encoder to learn both long-term dependencies and the distribution of gaussian noise $f(x_j^0|x_{1-j-1}^0, \epsilon_1^{s_1}, ..., \epsilon_{j-1}^{s_{j-1}})$. This improves the model's adaptability to inherent noise in downstream tasks. We are now working on a formal theoretical proof of combining autoregressive modeling with diffusion, which will be included in future work.
>
> **A4 for W2**
> Please refer to `a1 for q1`.
>
> Thank you for your encouraging score. We sincerely look forward to your reply.

---

> > ### Comment · Reviewer_baxN · 2025-04-04
> >
> > Thank you for addressing the key questions I raised. These supplementary clarifications have given me a more comprehensive understanding of the paper's value and significance. I will increase my rating for this paper.

---

> > > ### Author Response · Authors · 2025-04-04
> > >
> > > Dear Reviewer,
> > >
> > > Thank you for your constructive suggestions and positive feedback of our work. We sincerely appreciate your time and insightful comments. It is encouraging to hear that the supplementary addressed your concerns effectively.
> > > We are grateful for your willingness to reconsider the paper's rating, and we will carefully incorporate your suggestions in the future work.
> > >
> > > Best regards

---

### Decision · Program_Chairs · 2025-05-01

**Decision:**

Accept (poster)

**Comment:**

This paper proposed a self-supervised time series representation learning method. This research direction is significant and still in an early stages. The reviewers have generally provided positive feedback, and I also believe that this paper demonstrates good performance in both theoretical analysis and experimental validation. I agree with the comments of reviewer TNxq and suggest authors address the comments in the final revision. Overall, this paper presents a solid work and is potentially useful to the community. I am confident that this paper can be accepted.